# Hair follicle dermal condensation forms via Fgf20 primed cell cycle exit, cell motility, and aggregation

Leah C Biggs[†], Otto JM Mäkelä[†], Satu-Marja Myllymäki, Rishi Das Roy, Katja Närhi, Johanna Pispa, Tuija Mustonen, Marja L Mikkola*

Developmental Biology Program, Institute of Biotechnology, University of Helsinki, Helsinki, Finland

**Abstract** Mesenchymal condensation is a critical step in organogenesis, yet the underlying molecular and cellular mechanisms remain poorly understood. The hair follicle dermal condensate is the precursor to the permanent mesenchymal unit of the hair follicle, the dermal papilla, which regulates hair cycling throughout life and bears hair inductive potential. Dermal condensate morphogenesis depends on epithelial Fibroblast Growth Factor 20 (Fgf20). Here, we combine mouse models with 3D and 4D microscopy to demonstrate that dermal condensates form de novo and via directional migration. We identify cell cycle exit and cell shape changes as early hallmarks of dermal condensate morphogenesis and find that Fgf20 primes these cellular behaviors and enhances cell motility and condensation. RNAseq profiling of immediate Fgf20 targets revealed induction of a subset of dermal condensate marker genes. Collectively, these data indicate that dermal condensation occurs via directed cell movement and that Fgf20 orchestrates the early cellular and molecular events.
DOI: https://doi.org/10.7554/eLife.36468.001

**\*For correspondence:**
marja.mikkola@helsinki.fi

[†]These authors contributed equally to this work

**Competing interests:** The authors declare that no competing interests exist.

## Introduction

The mesenchymal condensation, first recognized in limb bud condensations and named 'precartilage condensates' by Dame Honor Fell (*Fell, 1925*), is a tissue-level structure preceding organ development. Since then, mesenchymal condensations have been described in the precursors of several organs, occurring in most ectodermal appendages (tooth, hair, mammary gland, feather, scales) as well as in bone and muscle (*Widelitz and Chuong, 1999*; *Hall and Miyake, 2000*; *Newman and Bhat, 2007*; *da Rocha-Azevedo and Grinnell, 2013*; *Biggs and Mikkola, 2014*). The condensation has been suggested to be the basic cellular unit of a tissue, and to function as the driver of morphogenesis (*Atchley and Hall, 1991*), yet the underlying molecular and cellular mechanisms remain largely unknown. Condensations are morphologically distinguishable and are defined as a local increase in cell density. Characteristics of condensing mesenchymal cells include a change in cell shape, close surface contact between adjacent cells, and increased nucleus-to-cytoplasm ratio (*Thorogood and Hinchliffe, 1975*; *Searls et al., 1972*) (for review see [*Hall and Miyake, 2000*]). Signals from the epithelium are critical for mesenchymal cell condensation. The question of how a local increase in cell density is achieved remains to be addressed. Several modes of cell condensation have been proposed, including i) increase in mitotic activity, ii) active migration of cells, and iii) failure of cells to disperse due to changes in cell-cell and/or cell-extracellular matrix (ECM) interactions (*Hall and Miyake, 1992*).

The hair follicle (HF) is an excellent model to study the early attributes of mesenchymal condensation. HFs of the mouse dorsum develop in three waves as a result of reciprocal epithelial-mesenchymal signaling events with the first subset initiating morphogenesis at embryonic day (E) E13.5 and

**eLife digest** All mammal hair springs from hair follicles under the skin. These follicles sit in the dermis, beneath the outermost skin layer, the epidermis. In the embryo, hair follicles develop from unspecialized cells in two tissues, the epithelium and the mesenchyme, which will later develop into the dermis and epidermis, respectively. As development progresses, the cells of these tissues begin to cluster, and signals passing back and forth between the epithelium and mesenchyme instruct the cells what to do. In the mesenchyme, cells called fibroblasts squeeze up against their neighbors, forming patches called dermal condensates. These mature into so-called dermal papillae, which supply specific molecules called growth factors that regulate hair formation throughout lifetime.

Fibroblasts in the developing skin respond to a signal from the epithelium called fibroblast growth factor 20 (Fgf20), but we do not yet understand its effects. It is possible that Fgf20 tells the cells to divide, forming clusters of daughter cells around their current location. Or, it could be that Fgf20 tells the cells to move, encouraging them to travel towards one another to form groups.

To address this question, Biggs, Mäkelä et al. examined developing mouse skin grown in the laboratory. They traced cells marked with fluorescent tags to analyze their behavior as the condensates formed. This revealed that the Fgf20 signal acts as a rallying call, triggering fibroblast movement. The cells changed shape and moved towards one another, rather than dividing to create their own clusters. In fact, they switched off their own cell cycle as the condensates formed, halting their ability to divide. A technique called RNA sequencing revealed that Fgf20 also promotes the use of genes known to be active in dermal condensates.

Dermal papillae control hair growth, and transplanting them under the skin can form new hair follicles. However, these cells lose this ability when grown in the laboratory. Understanding how they develop could be beneficial for future hair growth therapy. Further work could also address fundamental questions in embryology. Condensates of cells from the mesenchyme also precede the formation of limbs, bones, muscles and organs. Extending this work could help us to understand this critical developmental step.

DOI: https://doi.org/10.7554/eLife.36468.002

eventually producing guard hairs (*Hardy, 1992*). Within 24 hr, the previously homogenous epidermis exhibits discrete, focal thickenings, known as placodes, which are accompanied by condensation of the adjacent mesenchyme (*Biggs and Mikkola, 2014*). Coincident with dermal fibroblast condensation, their transcriptome profoundly alters. In particular, genes involved in cell-cell signaling such as Bmp4 and Wnt pathway components, p75 neurotrophin receptor, and many transcription factors including Sox2, one of the earliest markers of incipient dermal condensates (DC), are upregulated (*Driskell et al., 2009*; *Sennett et al., 2015*; *Jones et al., 1991*; *Botchkareva et al., 1999*). Another hallmark of DC formation is the differential expression of ECM molecules including tenascin, NCAM, and chondroitin sulfate proteoglycan, as well as syndecan-1 (*Richardson et al., 2009*). As HF morphogenesis continues, the DCs become enveloped by the down-growing follicular epithelium and subsequently mature into the permanent, mesenchymal component of the HF termed the dermal papilla (DP) (*Morgan, 2014*). The DP directs HF cycling throughout life and its miniaturization or absence results in a thinner or absent hair shaft (*Chi et al., 2013*; *Rompolas et al., 2012*). Of note, the DP has inductive capacity demonstrated by transplantation of freshly isolated or cultured (low passage) rodent DP cells under glabrous epithelium, which results in induction of hair follicle development (*Oliver, 1970*; *Jahoda et al., 1984*). By comparison, human DP cells lose their inductive capacity even faster in vitro but some activity can be sustained in 3D culture in aggregates (*Higgins et al., 2013*), suggesting that cellular condensation is tightly linked with DP cell fate specification and maintenance.

The molecular regulators of DC/DP morphogenesis have slowly begun to emerge. Absence of platelet-derived growth factor A (Pdgf-A) results in smaller DPs (*Karlsson et al., 1999*); however, the entire dermis is thinner, likely accounting for the DP phenotype as indicated by more recent studies in which the PDGF receptors were conditionally deleted in the dermis (*Rezza et al., 2015*). Another placode-derived factor implicated in DC formation is Sonic hedgehog (Shh) (*Karlsson et al., 1999*; *St-Jacques et al., 1998*; *Chiang et al., 1999*). However, conditional dermal deletion of Shh receptor

Smoothened indicates a role in DC maintenance rather than induction (*Woo et al., 2012*). Several other known DC markers such as Sox2 (*Clavel et al., 2012*), Tbx18 (*Grisanti et al., 2013a*), Cxcr4 (*Sennett et al., 2014*), and Enpp2/autotaxin (*Grisanti et al., 2013b*) have also been conditionally ablated, but none of them results in absence of the DC.

Perhaps surprisingly, the most informative genetic studies uncovering the molecular basis of DC formation have been those targeting the epithelium. Wnt signaling is believed to be the at the top of the hierarchy of signaling factors guiding HF morphogenesis, and expression of stabilized β-catenin in the epidermis results in broad adoption of placode fate as well as condensed mesenchyme concomitant with DC marker expression throughout the upper dermis (*Närhi et al., 2008*; *Zhang et al., 2008*; *Suzuki et al., 2009*). The ectodysplasin (Eda)/Edar pathway is another essential pathway for placode morphogenesis: in its absence only rudimentary primary hair placodes form transiently and DCs are missing (*Headon and Overbeek, 1999*; *Laurikkala et al., 2002*; *Schmidt-Ullrich et al., 2006*). Downstream of Wnt and Eda signaling, placodal factor Fgf20 is expressed early during HF morphogenesis (*Huh et al., 2013*; *Lefebvre et al., 2012*). Deletion of Fgf20 results in absent DC in guard hairs and many secondary (awl and auchene) hairs as shown by morphological and molecular analyses, as well as failure in placode invagination, and ultimately, absent hairs (*Huh et al., 2013*). Moreover, the condensed mesenchyme observed in embryos expressing stabilized epithelial β-catenin was ablated in the absence of Fgf20 further confirming the indispensable function of Fgf20 in DC induction.

Although these molecular players have been investigated, the cellular mechanism of DC development and the role that Fgf20 plays in DC morphogenesis remain unknown. We therefore aimed to define the hallmarks of DC morphogenesis and the function that Fgf20 plays in this process. We applied a multifaceted approach to determine the cellular and molecular changes leading to DC morphogenesis using murine back skin primary hair follicle as the model due to its feasibility for manipulation and ex vivo imaging (*Ahtiainen et al., 2014*). We identify cell shape changes and cycle exit as early hallmarks of DC morphogenesis. Live confocal imaging and lineage tracing showed that fibroblasts were recruited from the near vicinity and not from a pre-specified pool of Sox2+ cells, and migrate toward placodal epithelium. Further, Fgf20 induced fibroblast migration and cell shape change, as well as transcriptional responses that suggest its involvement in cell cycle exit. Collectively, our data show that Fgf20 governs multiple cellular and molecular events critical for DC formation.

## Results

### Dermal fibroblasts condense and change shape below the developing hair follicle placode

To determine to what extent the fibroblasts within the dermal condensate are more densely packed than the interfollicular fibroblasts, we examined primary HFs in E14.5 whole skin in 3D using confocal microscopy. The volume of the DC was determined by using a transgenic *Sox2-GFP* reporter (*Figure 1A*, left panel), whose expression correlates well with endogenous Sox2 ([*Driskell et al., 2009*]; *Figure 1A*), and the same volume was used on an interfollicular area of the upper dermis. Quantification of nuclei confirmed that cells were nearly twice as dense within the DC than in the interfollicular area (p=0.000106) (*Figure 1B*). We have previously shown that *Fgf20$^{β-Gal/β-Gal}$* (hereafter *Fgf20$^{-/-}$*) mice lack all molecular signs of primary DC formation including Sox2 (*Huh et al., 2013*) (*Figure 1A*, right panel), and quantification of dermal fibroblast density directly underneath the *Fgf20$^{-/-}$* placode revealed that these fibroblasts exhibit density similar to that of the wild-type interfollicular dermis (p=0.962425) (*Figure 1B*).

Next, we wanted to quantify DC formation in more detail. Primary hair placode induction is a continuous process first occurring near the mammary line and proceeding dorsally and caudally (*Dhouailly et al., 2004*), and hence the same embryo contains hair placodes in different developmental stages. To better capture the dynamic nature of DC formation, we categorized hair placodes in four developmental stages. Follicular epithelium was identified by the *Fgf20$^{β-gal}$* knock-in allele, one of the earliest placode markers ([*Huh et al., 2013*]; *Figure 1C*). Stage I was defined as a single layered placode, and stage II as a multilayered placode. Stage III is a placode that has invaginated into the dermis, and stage IV placodes have an anterior pocket where DC cells reside (*Figure 1C*).

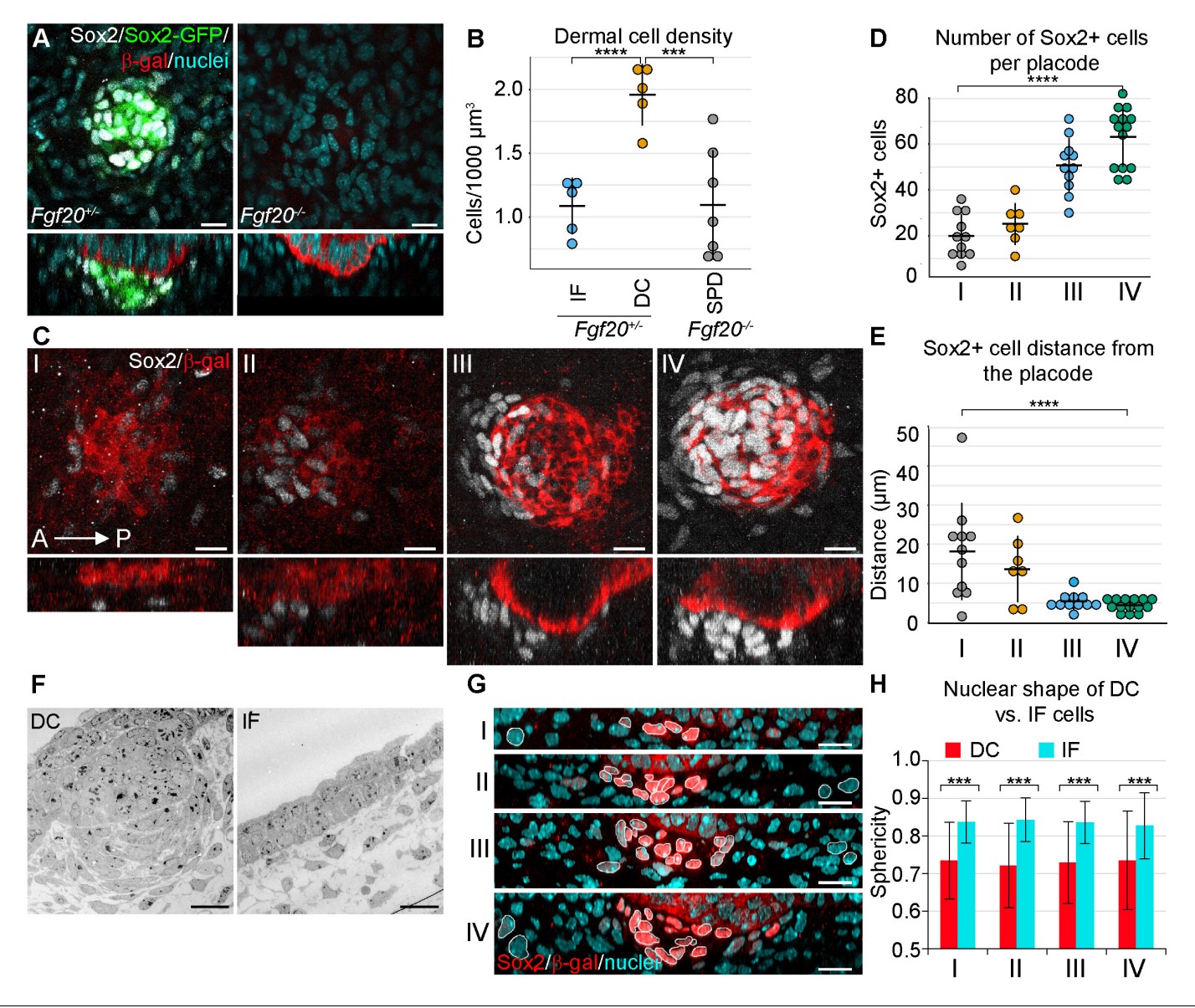

**Figure 1.** Changes in dermal cell characteristics upon dermal condensation induction. (**A**) Confocal microscopy immunofluorescent optical sections (planar and sagittal views) of E14.5 control (*Fgf20$^{+/-}$;Sox2-GFP*) and *Fgf20$^{-/-}$;Sox2-GFP* (green) embryonic skin labelled with antibodies against Sox2+ (white) and β-galactosidase (β-gal, red) to visualize DC and placodes, respectively. Note the absence of Sox2 antibody staining and Sox2-GFP reporter in *Fgf20$^{-/-}$* HFs. (**B**) Quantification of fibroblasts in E14.5 Sox2-GFP DC volume in *Fgf20$^{+/-}$* DC and interfolliclular upper dermis (IF) as well as in *Fgf20$^{-/-}$* dermis immediately adjacent to placodes (SPD), (*n* = 5 placodes from two skins Fgf20$^{+/-}$; *n* = 6 placodes from two skins *Fgf20$^{-/-}$*) unpaired Student's T-test. (**C**) Confocal microscopy immunofluorescent optical sections (planar and sagittal views) of *Fgf20$^{+/-}$* HF between E13.5 and E14.5, labeled with antibodies against Sox2 (white) and β-Gal (red). Placode morphogenesis was divided into four categories based on advancing development (I–IV). (**D**) Quantification of Sox2+ cells at each stage of placode morphogenesis (one-way ANOVA, *n* = 11, 7, 11, 13 placodes from 6, 4, 11, and 8 skins for stages I – IV, respectively). (**E**) Quantification of the median distance of Sox2+ cells to the nearest placode surface (one-way ANOVA, *n* = 11, 7, 11, 13 placodes from 6, 4, 11, and eight skins for stages I – IV, respectively) (**F**) Transmission electron micrographs of E14.5 wild-type skin dermal condensation (DC) and an interfollicular (IF) region. Note the convex nuclei and lack of space between the cells in the DC compared to the non-DC region. (**G**) Confocal optical sections (sagittal view) of advancing HF morphogenesis (stages I-IV); Sox2+ nuclei (red) are contrasted with Sox2- nuclei (blue); white outlines provide an example of cells compared. (**H**) Quantification of sphericity of Sox2+ and Sox2- nuclei; significance was assessed using Student's T-test (*n*$_{I}$ = 92 and 162 (6 placodes, three skins), *n*$_{II}$ = 253 and 163 (6 placodes, two skins), *n*$_{III}$ = 332 and 217 (6 placodes, two skins), *n*$_{IV}$ = 125 and 137 (8 placodes, three skins) DC and IF cells, respectively). A, anterior; P, posterior; SPD, sub-placodal dermis. Error bars represent standard deviation (SD). *p≤0.05; **p≤0.01; ***p≤0.001; ****p≤0.0001. Scale bar = 10 μm. See also *Figure 1—video 1* and *Figure 1—source data 1*.

DOI: https://doi.org/10.7554/eLife.36468.003

*Figure 1 continued on next page*

*Figure 1 continued*

The following video and source data are available for figure 1:

**Source data 1.** Values for quantification of morphological analysis of DC.
DOI: https://doi.org/10.7554/eLife.36468.004
**Figure 1—video 1.** Cell shapes of DC cells and non-DC fibroblasts at placode stage IV.
DOI: https://doi.org/10.7554/eLife.36468.005

The number and distance from placodes of Sox2+ cells was analyzed in 3D. Throughout these stages, the number of Sox2+ cells associated with each placode increased significantly (p<0.0001) (*Figure 1D*). Quantification of their distance from the placode revealed a progressive decrease in median distance (p<0.0001) (*Figure 1E*). At stage I, some dispersed Sox2+ cells were observed, which by stage II were preferentially oriented on the anterior side of the placode. By stage III, the cells appeared to be closer to the placode, and at stage IV, the Sox2+ cells maintained their proximity but increased in number (*Figure 1D,E*).

Mesenchymal cell condensation is often accompanied by cell shape changes (*Ray and Chapman, 2015*; *Mammoto et al., 2011*). Our 3D analysis (*Figure 1A,C*) also suggested that DC formation correlates with cell shape changes. Transmission electron microscopy (TEM) revealed that E14.5 DC cells have an elongated, convex shape and display a characteristic alignment next to each other (*Figure 1F*). 3D rendering of nuclear shapes in whole mount and subsequent quantifications of nuclear sphericity during DC formation showed that the change in nuclear shape is an early indicator of DC formation (all stages p<0.0001) (*Figure 1G,H*). Consistent with the TEM and nuclear data, 3D rendering of cell shapes based on a ubiquitous cell membrane marker confirmed that Sox2+ DC cells exhibited a convex shape, whereas the non-DC fibroblasts were relatively spherical (*Figure 1—video 1*).

## Dermal condensate cells acquire Sox2 expression de novo

Given that the number of Sox2+ cells increased while their distance to placode decreased over time, and that the dermis contains Sox2+ Schwann cell precursors (*Jessen and Mirsky, 2005*), we next asked whether the DC cells were recruited from a pre-existing pool of Sox2+ cells or whether they acquired Sox2 expression de novo. To this end, we utilized $Sox2^{creERT2}$;$R26R^{tdTomato/+}$ embryos and analyzed tdTomato expression after 24 or 48 hr of tamoxifen (TAM) exposure beginning at E12.5 (before morphological and molecular appearance of HF), E13.5 (earliest molecular sign of HF), and at E14.5 (placode stage of HF) (*Figure 2A–F*). To minimize the effect of variation in HF development, we examined hair follicles from the same region in all embryos (ventro-lateral skin) and quantified the total number of Sox2+ cells using Sox2 antibody and analyzed the proportion of tdTomato-labeled cells amongst them (*Figure 2G,H*). When labeling was induced at E14.5 and cells analyzed 24 hr later, 65% of Sox2+ cells were tdTomato+, indicating that the TAM dosage used results in a relatively high labeling efficiency (*Figure 2D,G*). Strikingly, when TAM was administered at E13.5, only 20% of Sox2+ cells were tdTomato 24 hr later (*Figure 2C,G*). This proportion, however, increased substantially when mice were analyzed at E15.5 (*Figure 2F,G*). Finally, when TAM was administered at E12.5 and DC cells analyzed at E13.5 and 14.5, only 5% and 8% of Sox2+ cells were tdTomato+, respectively (*Figure 2B,E,G*) indicating that there is no pre-specified pool of Sox2+ cells. Analysis of secondary hair placodes (that form at E15.5) in mice injected with TAM at E13.5 or E14.5 revealed that 5% and 12% of Sox2+ cells were labeled, respectively, further confirming that DC cells acquire Sox2 expression de novo (*Figure 2—figure supplement 1*).

While quantifying the Sox2+;tdTomato+ cells in the DC, we noticed that they exhibited a preferential location adjacent to the placode, while cells that recently acquired DC fate (Sox2+, tdTomato-) appeared further away from the epithelium (*Figure 2*). We quantified this phenomenon in E15.5 HFs labeled at E14.5 (where 65% of the Sox2+ were tdTomato+) (*Figure 2G,H*) and measured the median distance of the tdTomato+ cells to the center of the placode surface. Our analysis revealed that the Sox2+;tdTomato+ cells were significantly closer to the placode than Sox2+;tdTomato- cells (p=0.0105) (*Figure 2I*). Nearest neighbor analysis showed that 87% of tdTomato positive cells had a tdTomato+ neighbor and hence were not randomly distributed (p<0.0001) (*Figure 2J*). Together, these data indicate that DC cells gain Sox2 expression just prior to becoming a DC cell. Further, we

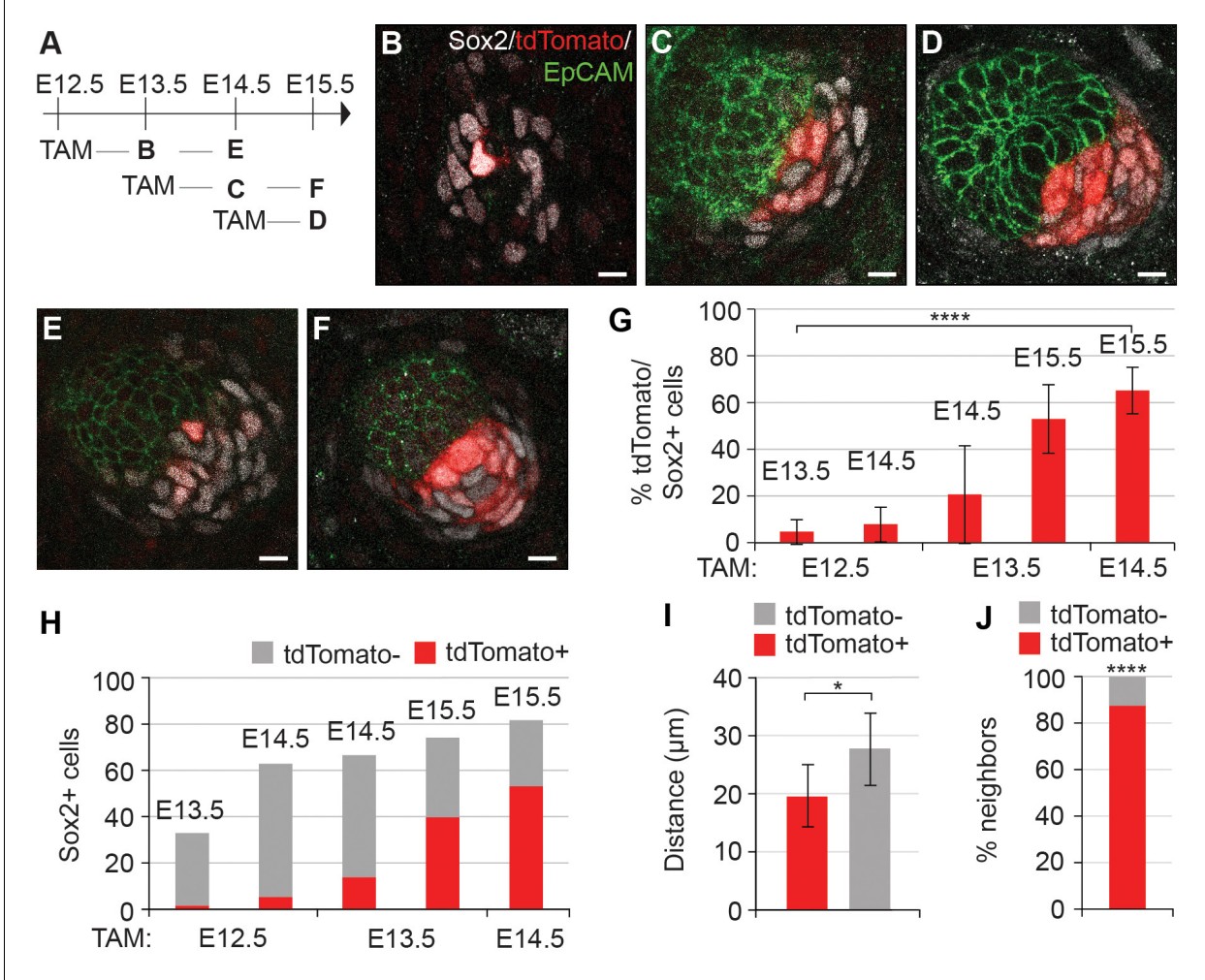

**Figure 2.** Dermal condensation cells acquire Sox2 expression de novo and populate the DC upon Sox2 acquisition. (A) Scheme of tamoxifen (TAM) injection and analysis. A single labeling dose of TAM was administered to pregnant dams at E12.5, E13.5, or E14.5. (B–F) Confocal microscopy immunofluorescent optical sections (planar view) of *Sox2^{creERT};R26R^{tdTomato}* skins immunolabeled with Sox2 (white), tdTomato label (red), and EpCAM (green) at 24 or 48 hr after TAM administration (n > 4 injections; E12.5 + 24 hr, n = 23 DCs from nine skins; E12.5 + 48 hr, n = 23 DCs from seven skins; E13.5 + 24 hr, n = 33 DCs from 11 skins; E13.5 + 48 hr, n = 23 DCs from eight skins; E14.5 + 24 hr, n = 21 DCs from nine skins). (B) E12.5 + 24 hr resulted in very few labeled Sox2 cells. (C) E13.5 + 24 hr showed increased labeling of Sox2+ cells. (D) TAM induction at E14.5 resulted in a majority of Sox2+ cells labeled within 24 hr. (E) E12.5 + 48 hr resulted in a low number of tdTomato+, Sox2+ cells. (F) E13.5 + 48 hr resulted in a large number of Sox2+ cells labeled with tdTomato. (G) Quantification of Sox2+ cells positive for tdTomato label at indicated time points; significance was assessed with one-way ANOVA. (H) Quantification of the average number of tdTomato+ cells as part of the whole Sox2+ cell population (E12.5 + 24 hr=2 of 33 cells; E12.5 + 48 hr=5 of 63 cells; E13.5 + 24 hr=14 of 67 cells; E13.5 + 48 hr=40 of 74 cells; E14.5 + 24 hr=53 of 82 cells). (I) Quantification of the distance of tdTomato+ and tdTomato- (n = 11 placodes from four skins) Sox2+ cells from placode surface, significance was assessed with Student's T-test. (J) Nearest neighbor analysis of Sox2+ cells for tdTomato label (87.7%) vs unlabeled (12.2%) at E15.5 (n = 635), significance was assessed with Chi-square test. Error bars represent SD. *p≤0.05; ****p≤0.0001. Scale bar = 10 μm. See also *Figure 2—source data 1* and *Figure 2—figure supplement 1*.

DOI: https://doi.org/10.7554/eLife.36468.006

The following source data and figure supplements are available for figure 2:

**Source data 1.** Values for quantification of Sox2 lineage tracing in primary placodes.

DOI: https://doi.org/10.7554/eLife.36468.009

**Figure supplement 1.** Secondary HF DC gain *Sox2* expression de novo.

DOI: https://doi.org/10.7554/eLife.36468.007

**Figure supplement 1—source data 1.** Values for quantification of Sox2 lineage tracing in secondary placodes.

DOI: https://doi.org/10.7554/eLife.36468.008

find that cells do not randomly assort after entering the DC, thus the 'oldest' DC cells are most likely to be located closest to the placode.

## Dermal condensate formation is associated with cell cycle exit

An increase in cell number can be achieved in a number of ways. Our quantifications revealed that when TAM was administered at E13.5, on average 20% and 53% of Sox2+ cells were tdTomato+ at E14.5 and 15.5, respectively (*Figure 2G*). Given that the average number of Sox2+ DC cells increased only by seven cells (from 67 to 74) from E14.5 to E15.5 in the same dataset (*Figure 2H*), our findings suggest that either TAM remains active for longer than 24 hr as previously suggested (*Hayashi and McMahon, 2002*), or that Sox2+ cells labeled between E13.5 and E14.5 increase in number by proliferation. To test whether locally enhanced proliferation could drive DC formation, we assessed cell cycle dynamics during DC morphogenesis with the aid of a bitransgenic cell cycle indicator Fucci mouse model (*Sakaue-Sawano et al., 2008*) in which mKusabira Orange (mKO) is expressed during the $G_1/G_0$ phase (hereafter $G_1$) and mAzami Green (mAG) during the S/$G_2$/M phase (*Figure 3A*). The interfollicular dermal cells (Sox2-) were equally distributed between $G_1$ and S/$G_2$/M phases at all stages of placode morphogenesis analyzed (*Figure 3B*). In contrast, the Sox2 + cells showed progressive exit from the cell cycle. At stage I, the Sox2+ cells were nearly evenly distributed between proliferative and non-proliferative phases. By stage II, the majority of Sox2+ cells were in $G_1$, which persisted through stages III and IV ($p_I$ = 0.0166, $p_{II}$ = 0.002, $p_{III, IV}$ < 0.001, *Figure 3B*). Further, the percent non-proliferating cells in the stage I DC was significantly lower than the following stages ($p_{IvsII}$ = 0.0211), suggesting that DC fate acquisition occurs before cell cycle exit. This cell cycle exit was not transient as the vast majority of DC cells remained in $G_1$ through E15.5 (*Figure 3B,C*). To determine whether this cell cycle exit is dependent on Fgf20, we analyzed the cell cycle status of E14.5 *Fgf20*$^{-/-}$ fibroblasts. Immediately below the placode, in a volume equivalent to a wild-type stage IV DC, the proportion of $G_1$ and S/$G_2$/M cells did not differ from that of the interfollicular dermal cells (p=0.473, *Figure 3B and D*). To further substantiate our findings, we examined the cell cycle distribution in mice overexpressing Eda under the *Keratin14* promoter (*K14-Eda*), a model of enlarged DC. Not only are the placodes bigger at E14.5 than in the wild type (*Mustonen et al., 2004*; *Ahtiainen et al., 2014*), but there are also more Sox2+ cells (*Huh et al., 2013*). Similar to control DC, about 95% of the Sox2+ DC cells were in the $G_1$ phase (*Figure 3B,E*) indicating that increased DC size can be achieved without enhancing cellular proliferation. Similar findings were observed with the *R26Fucci2aR* cell cycle indicator mouse model (*Figure 3—figure supplement 1*).

To confirm our observations, we further assessed cell proliferation using a uridine-analogue EdU incorporation assay at different stages of DC development (*Figure 3F*). Concordantly with the Fucci-reporter analysis, at stage I we observed that 18% of the Sox2+ DC cells were EdU positive, but at stage II the proportion of EdU-positive cells dropped significantly to 5.2% ($p_{IvsII}$ = 0.003) and remained unchanged through stage IV ($p_{IIvsIII}$ = 0.277, $p_{IIIvsIV}$ = 0.591). Collectively, these data suggest that cell cycle exit is a hallmark of dermal condensate morphogenesis.

## Directed migration of fibroblasts drives dermal condensate formation

Having ruled out proliferation as a mechanism for the increased cell density in DC cells, we next assessed the contribution of cellular migration by using live, confocal 3D imaging. We utilized *Sox2-GFP;Fucci*$^{mKO}$ skins to monitor the behavior of DC-forming cells and as a control, tracked interfollicular, non-DC fibroblasts (Sox2-GFP−) at the corresponding cell cycle phase (Fucci$^{mKO}$+ cells). Cultures were initiated at E13.75, a time point when incipient DCs were visible (*Figure 4A* and *Video 1*).

To characterize cell movement in detail, we quantified velocity, net velocity, straightness, and directionality. We initially tracked all Sox2-GFP+ DC cells (*Figure 4A*−C), but observed that many of the cells that were present in the condensates at the beginning of tracing displayed low velocity suggesting that their movement is restricted (*Figure 4D*). However, cells that were initially further than 30 μm (the average DC diameter) from the center of the condensate at the beginning of tracing behaved differently. Notably, these cells showed a preferential movement direction towards the condensate center, while interfollicular cells exhibited no directionality (*Figure 4B,C*). We observed that the condensate-forming fibroblasts also moved at a significantly higher velocity than interfollicular

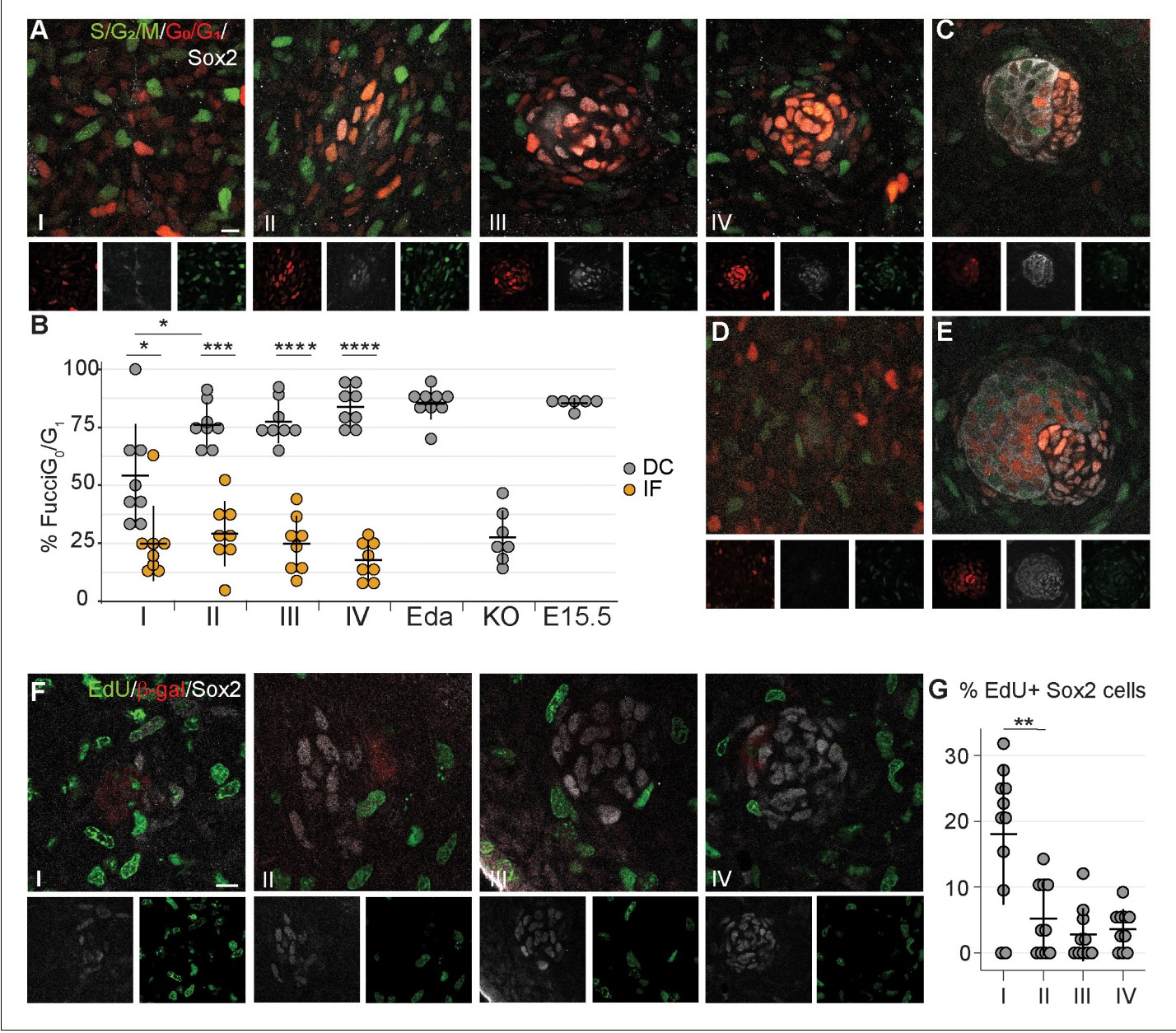

**Figure 3.** Dermal condensation cells exhibit cell cycle exit. (**A, C–E**) Confocal microscopy optical planar sections of *Fgf20$^{+/-}$;Fucci* (G$_1$ red; S/G$_2$/M green) skins at indicated stages of HF placode morphogenesis (**I–IV**) were labeled with antibodies against β-Gal (cytoplasmic white) and Sox2 (nuclear white). (**A**) The Sox2+ nuclei were scored as red, green, both, or neither and compared to the interfollicular Sox2- fibroblasts (n = eight placodes per stage from seven, four, five, and five skins in stages I, II, III, and IV, respectively). (**B**) Quantification of percent Sox2+ (DC) cells and Sox2- (IF) fibroblasts in G$_0$/G$_1$ phase during HF placode morphogenesis (**I–IV**), in E14.5 *K14-Eda* (Eda), in E14.5 *Fgf20$^{-/-}$* (KO), and in E15.5 *Fgf20$^{+/-}$* (E15.5), paired Student's T-test. (**C–E**) Expression of Fucci transgenes in (**C**) E15.5 control DCs (n = nine placodes from four skins), (**D**) E14.5 *Fgf20$^{-/-}$* dermis immediately adjacent to the placode (n = seven placodes from three skins), and (**E**) E14.5 *K14-Eda* (n = six placodes from two skins) DCs. (**F**) Confocal microscopy optical planar sections of *Fgf20$^{+/-}$* skins at indicated stages of HF placode morphogenesis (**I–IV**). Embryos were subjected to 2 hr EdU pulse in utero prior to sacrifice. Skins were treated with Click-It detection to visualize EdU-positive cells (green) and immunolabeled with Sox2 (white) and β-gal (red, not shown). (**G**) Quantification of EdU-positive Sox2 DC cells (n$_I$ = 11 placodes from three skins, n$_{II}$ = 10 placodes from five skins, n$_{III}$ = 10 placodes from five skins, n$_{IV}$ = 11 placodes from five skins). *, p≤0.05; ***, p≤0.001 ****, p≤0.0001. Error bars represent SD. Scale bar = 10 μm. See also **Figure 3—source data 1**.

DOI: https://doi.org/10.7554/eLife.36468.010

The following source data and figure supplement are available for figure 3:

**Source data 1.** Values for quantification of cell cycle analysis during DC morphogenesis.

*Figure 3 continued on next page*

*Figure 3 continued*

DOI: https://doi.org/10.7554/eLife.36468.012

**Figure supplement 1.** *R26Fucci2aR* expression during DC morphogenesis.

DOI: https://doi.org/10.7554/eLife.36468.011

cells (p=0.0051) (*Figure 4D*) yet there was no significant difference between the straightness of cell movement or net velocity (p=0.8945 and p=0.1390, respectively) (*Figure 4E,F*). Furthermore, when we analyzed migration of condensate-forming fibroblasts only until they enter the dermal condensate, not only were they preferentially moving toward the DC center but also migrated on a significantly straighter track (p=0.0232) and had higher net velocity than the interfollicular cells (p=0.0006) (*Figure 4—figure supplement 1*).

As cell migration is associated with remodeling of the actin cytoskeleton (for review see [*Svitkina, 2018*]), we investigated whether Sox2-GFP cells presumably *en route* to the DC could be distinguished from other dermal fibroblasts or Sox2-GFP cells already present in the DC based on the organization of their actin cytoskeleton. The non-DC Sox2-GFP cells displayed a slightly higher intensity of F-actin than dermal fibroblasts, which may be indicative of their migratory status (*Figure 4—figure supplement 2A,B*). However, the Sox2-GFP cells within the DC displayed even higher levels of F-actin intensity compared to non-DC Sox2-GFP cells (*Figure 4—figure supplement 2A,B*). Further, the intensity of F-actin increased as the DC progressed through morphogenesis (*Figure 4—figure supplement 2D,E*). As these cells exhibited reduced motility, this suggests that the F-actin is involved in maintaining the 3D configuration of the DC. Together, these results suggest that directed migration of the dermal fibroblasts is driving DC formation, but once the cells have entered the DC, their motile behavior changes, likely due to limitations posed by increased cell density, a finding in line with our lineage tracing/nearest neighbor analysis (*Figure 2I,J*).

## Transcriptional responses of Fgf20 in the dermis

In order to address the function of Fgf20 in DC formation, we first aimed to identify its immediate transcriptional targets. Isolated E13.5 *Fgf20*$^{-/-}$ dermises were cut into two pieces: one half was incubated in recombinant FGF20 for 3 hr, and the other half in BSA. RNA sequencing was carried out on five pairs of biological replicates (*Figure 5A*). Differential gene expression analysis revealed 40 protein-coding genes including many known Fgf/MAPK pathway target genes and feedback regulators (*Ornitz and Itoh, 2015*; *Murphy et al., 2010*), such as those within the Dual-specificity phosphatase (Dusp), Sprouty and Sprouty-related Spred gene families (*Table 1*). We validated the RNAseq by qRT-PCR analysis of a subset of these genes in independently generated samples (*Figure 5B*). Of the 31 upregulated genes, 7 belong to the recently identified DC signature, and an additional 5 genes were >2 x more highly expressed in DC compared to non-DC fibroblasts in the same study (*Sennett et al., 2015*). Notably, several genes implicated in cell cycle regulation such as *Bcl6* and *Cdkn1a* (*p21*), a cyclin-dependent kinase inhibitor, which we previously identified as an early DC marker (*Huh et al., 2013*), were among the upregulated genes (*Table 1*).

To assess whether Fgf20 could also drive the expression of the potential transcriptional target genes in vivo we attempted to create a gain-of-function mouse line expressing Fgf20 under the *K14* promoter. Unfortunately, we were unsuccessful in generating *K14-Fgf20* lines that would display detectable *Fgf20* overexpression, and we therefore took advantage of a mouse model overexpressing Edar, the upstream regulator of Fgf20, under the *K14* promoter (*Pispa et al., 2004*). We confirmed upregulation of *Edar* expression in the entire basal epithelium by in situ hybridization (*Figure 5—figure supplement 1*). Accordingly, Fgf20, visualized by the *Fgf20*$^{\beta\text{-}gal}$ knock-in reporter, was ectopically expressed in the basal layer from E15.5 onwards (*Huh et al., 2013*) (*Figure 5C*). Consistent with our RNAseq analysis, we observed ectopic *Cdkn1a* expression in the mesenchyme immediately adjacent to the *Fgf20*-expressing epithelium on the *K14-Edar* background, whereas in the control dermis its expression was confined to the DC (*Figure 5C*), as reported previously (*Huh et al., 2013*). Importantly, *Cdkn1a* upregulation was Fgf20-dependent, as shown by its absence on the *K14-Edar;Fgf20*$^{-/-}$ background (*Figure 5C*). Further, *Sox2* which was not upregulated by Fgf20 in our RNAseq data, was not detected in the *K14-Edar* or *K14-Edar;Fgf20*$^{-/-}$ embryos, but was readily observed in control embryos (*Figure 5C* and *Figure 5—figure supplement 1*). Collectively, these

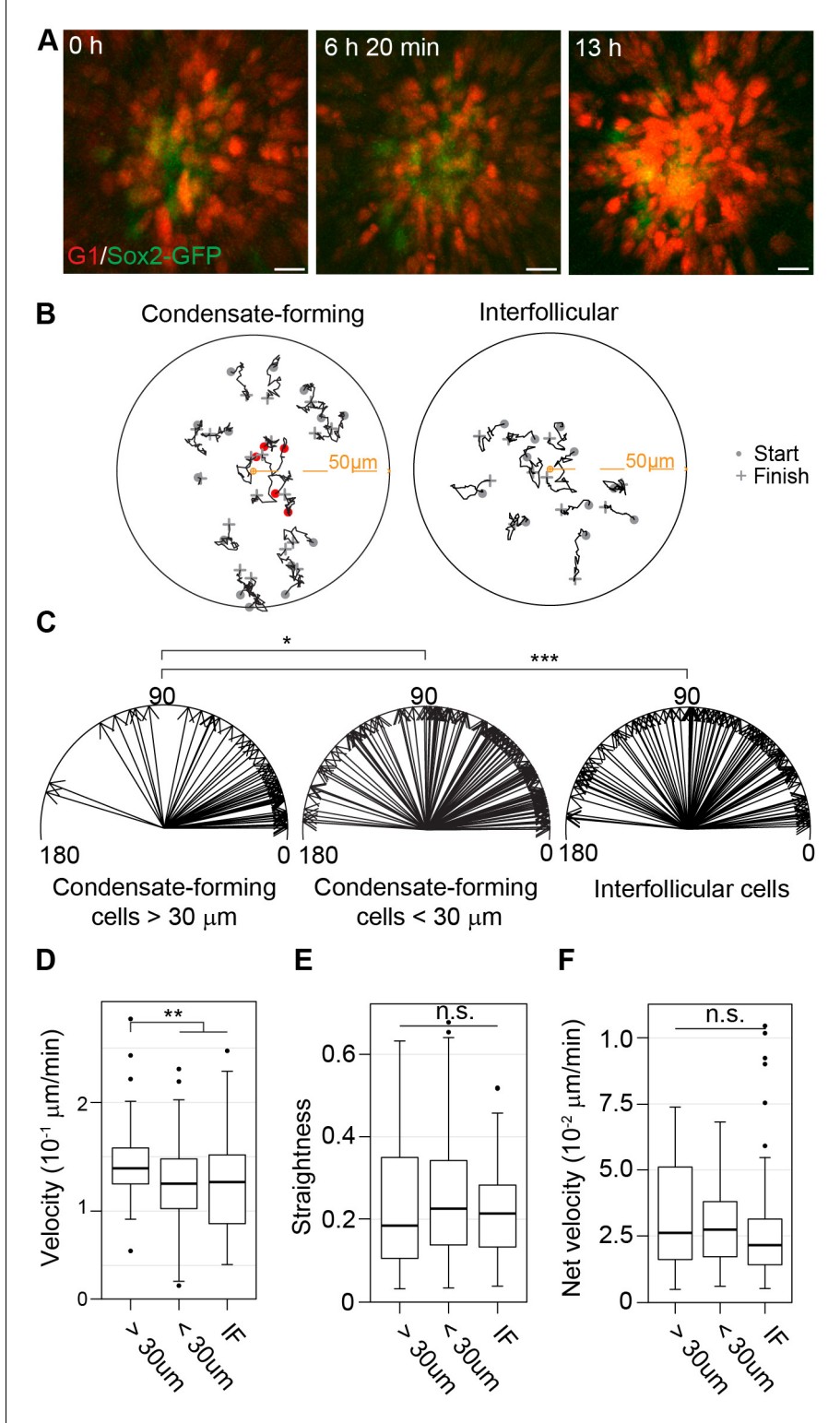

**Figure 4.** Dermal condensate formation is driven by directed migration of dermal fibroblasts. (**A**) Maximum intensity projections of indicated time points from live confocal imaging of E13.75 *Sox2-GFP; Fucci^mKO* skins. (**B**) Representative 2D plots of movement tracks of condensate forming (left) or IF (right) cells that are initially >30 μm (grey) and <30 μm (red) from the DC center. (**C**) Vectors of escape angles (the angle between cell trajectory in respect to center of the DC/interfollicular area and endpoint of trajectory) of cells that were initially >30 μm (left, *n* = 49 cells from eight placodes, images from six skin explants) and <30 μm (center, *n* = 127 cells from eight

*Figure 4 continued on next page*

*Figure 4 continued*

placodes, images from six skin explants) from the DC center, and interfollicular cells (right, *n* = 97 cells from eight placodes, images from six skin explants). Condensate-forming cells initially >30 µm from condensate center preferentially migrate toward condensate center (median 22°) whereas condensate-forming cells initially <30 µm from the center (median 48°) and interfollicular (median 68°) cells show no preferential direction of movement. Watson's U2 test shows a significant difference in escape angles in DC cells initially further away (>30 µm) from DC center versus interfollicular cells (p<0.001) or DC cells initially close (<30 µm) to the DC center (p<0.05). (D–F) Distribution of (D) cell velocity, (E) straightness, and (F) net velocity during DC formation. Significance was assessed with Mann-Whitney test. Condensate forming cells migrating initially >30 µm away from the DC center migrate faster than condensate-forming cells initially close (<30 µm) to the DC center (p=0.0022) and interfollicular cells (0.0051), but no difference was observed in track straightness (p=0.8945 and p=0.2376, respectively) or net velocity (p=0.5949 and p=0.139, respectively) between the groups. n.s., not significant; *p≤0.05; **p≤0.01; ***p≤0.001. Error bars represent SD. See also *Video 1*, *Figure 4—source data 1*, and *Figure 4—figure supplement 1*.

DOI: https://doi.org/10.7554/eLife.36468.013

The following source data and figure supplements are available for figure 4:

**Source data 1.** Values used for quantification of Sox2+ cell and IF fibroblast movement.
DOI: https://doi.org/10.7554/eLife.36468.018
**Figure supplement 1.** Condensate forming cells show directed migration before DC entry.
DOI: https://doi.org/10.7554/eLife.36468.014
**Figure supplement 1—source data 1.** Values used for quantification of Sox2+ cell movement until entry into DC.
DOI: https://doi.org/10.7554/eLife.36468.015
**Figure supplement 2.** Condensate-forming cells display elevated levels of F-actin.
DOI: https://doi.org/10.7554/eLife.36468.016
**Figure supplement 2—source data 1.** Values used to quantify phalloidin intensity.
DOI: https://doi.org/10.7554/eLife.36468.017

---

data suggest that Fgf20 regulates a subset of DC genes including *Cdkn1a*, a well-characterized inducer of $G_1$ arrest of the cell cycle.

## Dermal fibroblasts migrate in vitro and condense ex vivo in response to Fgf20

Our analyses of 3D live imaging data showed that directional migration drives dermal condensate morphogenesis. This observation led us to ask whether Fgf20 signaling plays a role in this process. First, we analyzed the effect of Fgf20 signaling on migration of a monolayer of growth-arrested E13.5 primary dermal fibroblasts in a scratch wound healing assay over 24 hr. FGF20 induced significantly faster wound closure compared to control fibroblasts (p=0.0177), an effect that was abolished in the presence of an Fgfr inhibitor, SU5402 (p=0.6100), confirming the specificity of the response to FGF20 (*Figure 6A and B*). SU5402 alone had no significant effect on wound closure (p=0.2056) (*Figure 6A and B*). Similar observations were made when cells were treated with FGF9, a member of the same Fgf subfamily as Fgf20 (*Figure 6—figure supplement 1*).

The scratch wound healing assay, however, does not distinguish between directional (chemotaxis) and non-directional cell migration (chemokinesis). To assess whether Fgf20 could function as a chemotactic factor, we analyzed migration of growth-arrested E13.5 primary dermal fibroblasts in a transwell assay. When FGF20 was present in the lower chamber only, the number of migrating

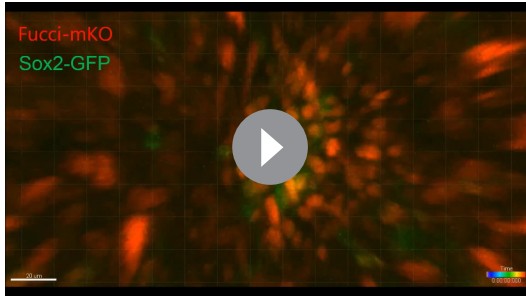

**Video 1.** Time-lapse fluorescence confocal video of DC morphogenesis. Dorsal skin from E13.75 *Fucci-mKO; Sox2-GFP* was explanted into Trowell culture set up and imaged with Leica TCS SP5 confocal microscope for 13 hr. Tracks of manually traced DC cells (Sox2-GFP +, green; Fucci-mKO+, red) and non-DC fibroblasts (Sox2-GFP-; Fucci-mKO+, red cells) are shown. See also *Figure 4* and *Figure 4—figure supplement 1*.
DOI: https://doi.org/10.7554/eLife.36468.019

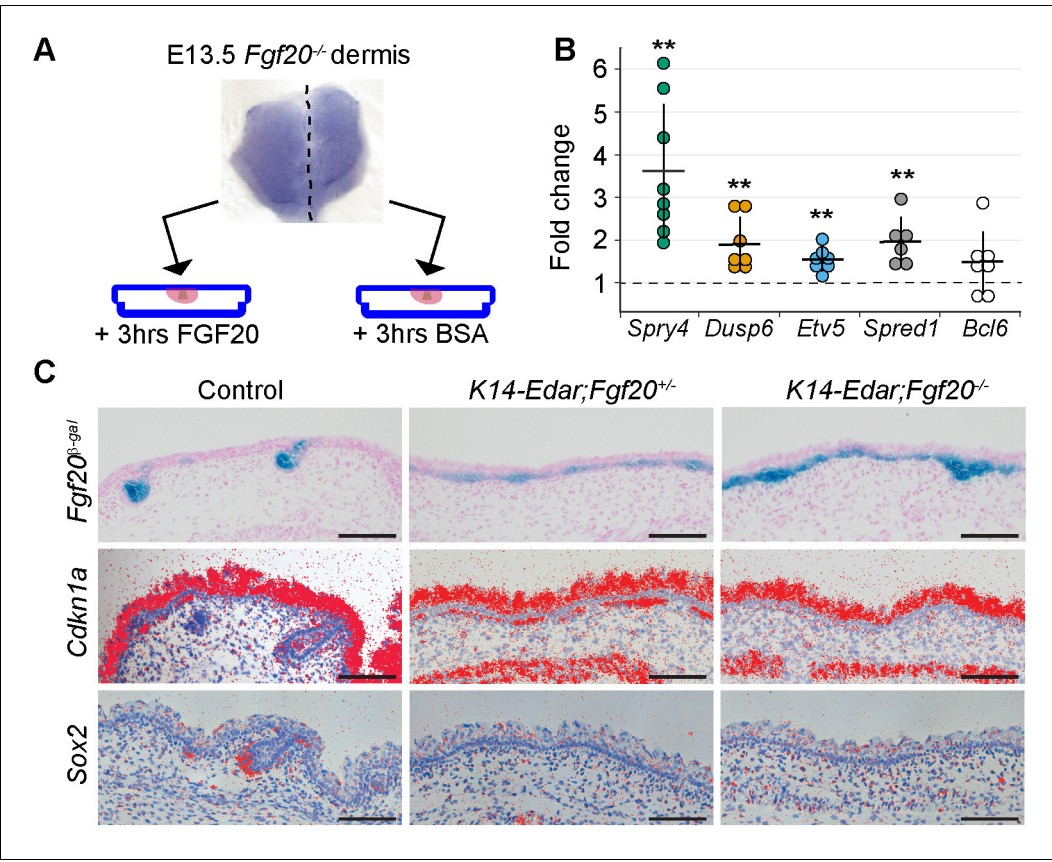

**Figure 5.** Fgf20 transcriptional targets in the dermis. (**A**) Schematic of the experimental setup. E13.5 *Fgf20*$^{-/-}$ dermises were separated into halves along the dorsal midline, each half was cultured for 3 hr in the presence of either 1 μg/ml FGF20 or with 0.1% BSA vehicle control. RNA was extracted and processed for RNA sequencing. (**B**) qRT-PCR was carried out on replicate samples for *Spry4* (*n* = 8), *Dusp6* (*n* = 7), *Etv5* (*n* = 7), *Spred1* (*n* = 6), and *Bcl6* (*n* = 7). Significance was assessed with one-sample T-test, \*\*=p < 0.01. Error bars represent SD. (**C**) Skins from E15.5 *Fgf20*$^{+/-}$ (control), *K14-Edar;Fgf20*$^{+/-}$, and *K14-Edar;Fgf20*$^{-/-}$ embryos (*n* = 6 embryos each) were assayed for β-galactosidase activity to assess the expression of the *Fgf20*$^{β-Gal}$ knock-in allele (top). Note the follicular localization of β-Gal activity in Fgf20$^{+/-}$ embryos, which is localized throughout the epidermis in *K14-Edar;Fgf20*$^{+/-}$, and *K14-Edar;Fgf20*$^{-/-}$ embryos. Radioactive in situ hybridization was utilized to detect *Cdkn1a* (*p21*) (middle) and *Sox2* (bottom) at E16.5. Note that *Cdkn1a* was restricted to the DC in the dermis in control embryos, but was localized throughout the upper dermis in *K14-Edar* embryos in an Fgf20-dependent manner. *Cdkn1a* was also strongly expressed in the differentiating epidermis and the *panniculus carnosus* muscle. See also *Table 1*, *Figure 5—source data*, and *Figure 5—figure supplement 1*.

DOI: https://doi.org/10.7554/eLife.36468.020

The following source data and figure supplements are available for figure 5:

**Source data 1.** Values used for qRT-PCR analysis of FGF20-treated Fgf20$^{-/-}$ dermis.
DOI: https://doi.org/10.7554/eLife.36468.023

**Figure supplement 1.** Analysis of Edar expression and dermal cell density in control, *K14-Edar;Fgf20*$^{+/-}$ skins *K14-Edar;Fgf20*$^{-/-}$ embryos.
DOI: https://doi.org/10.7554/eLife.36468.021

**Figure supplement 1—source data 1.** Values used to quantify fibroblast density in dermis.
DOI: https://doi.org/10.7554/eLife.36468.022

cells was significantly higher than in control wells (p=0.0003) (**Figure 6C**). When the FGF20 gradient was abolished by adding FGF20 also to the upper chamber, again a significantly higher number of cells migrated to the lower chamber (p<0.0001), however, there was no difference between the two types of FGF20 treatments (p=0.2256) (**Figure 6C**). Similar data was obtained when FGF9 was used

**Table 1.** Differentially expressed genes after 3 hr FGF20 treatment.

Genes in red: Fold DC vs. Fb is >2 x in *Sennett et al., 2015*; * indicates a DC signature gene. Genes in blue: Fold Fb vs. DC is >2 x in *Sennett et al., 2015*; # indicates a Fibroblast signature gene. See also *Figure 5* and *Figure 5—figure supplement 1*.

| Ensembl gene ID | Gene symbol | Log2 fold change | q-value |
|---|---|---|---|
| ENSMUSG00000024427 | Spry4 | 1,24306661 | 2,87958E-14 |
| ENSMUSG00000040276 | Pacsin1 | 1,117757314 | 0,005409828 |
| ENSMUSG00000000938 | Hoxa10 | 1,067306677 | 4,89962E-05 |
| ENSMUSG00000022484 | Hoxc10 | 1,055122804 | 0,000135006 |
| ENSMUSG00000037580 | Gch1 | 0,953581456 | 0,001623219 |
| ENSMUSG00000039628 | Hs3st6* | 0,9259377 | 0,048932785 |
| ENSMUSG00000013089 | Etv5 | 0,904428558 | 3,51931E-06 |
| ENSMUSG00000019960 | Dusp6 | 0,768627588 | 2,59873E-05 |
| ENSMUSG00000000435 | Myf5 | 0,741106812 | 0,021043062 |
| ENSMUSG00000022508 | Bcl6* | 0,71492512 | 0,001178264 |
| ENSMUSG00000014813 | Stc1 | 0,70075808 | 8,12534E-07 |
| ENSMUSG00000022367 | Has2 | 0,688772195 | 0,000329738 |
| ENSMUSG00000046223 | Plaur | 0,680287177 | 0,019395279 |
| ENSMUSG00000037211 | Spry1* | 0,668125166 | 0,000252705 |
| ENSMUSG00000020023 | Tmcc3 | 0,652897112 | 0,017991893 |
| ENSMUSG00000045671 | Spred2 | 0,646652707 | 0,021043062 |
| ENSMUSG00000032020 | Ubash3b* | 0,608057108 | 0,025457783 |
| ENSMUSG00000022114 | Spry2# | 0,604959493 | 3,29426E-05 |
| ENSMUSG00000039680 | Mrps6 | 0,587495288 | 0,048932785 |
| ENSMUSG00000043099 | Hic1 | 0,58156032 | 7,06147E-06 |
| ENSMUSG00000021567 | Nkd2 | 0,574323421 | 0,048932785 |
| ENSMUSG00000025402 | Nab2 | 0,573087974 | 0,001623219 |
| ENSMUSG00000026064 | Ptp4a1 | 0,560735472 | 0,049490799 |
| ENSMUSG00000026655 | Fam107b* | 0,552834404 | 0,000598789 |
| ENSMUSG00000046324 | Ermp1 | 0,545018197 | 0,004713961 |
| ENSMUSG00000015957 | Wnt11# | 0,521839644 | 0,030834254 |
| ENSMUSG00000023067 | Cdkn1a* | 0,509994166 | 0,048932785 |
| ENSMUSG00000027351 | Spred1* | 0,501328957 | 0,017546304 |
| ENSMUSG00000053716 | Dusp7 | 0,498184179 | 0,017991893 |
| ENSMUSG00000007029 | Vars | 0,396158628 | 0,048932785 |
| ENSMUSG00000018001 | Cyth3 | 0,388676591 | 0,017546304 |
| ENSMUSG00000029563 | Foxp2 | −0,555487269 | 0,028254118 |
| ENSMUSG00000046743 | Fat4# | −0,569844598 | 0,006906967 |
| ENSMUSG00000036995 | Asap3 | −0,763477567 | 0,021457852 |
| ENSMUSG00000028036 | Ptgfr# | −0,85391079 | 6,19325E-06 |
| ENSMUSG00000035352 | Ccl12 | −0,855627781 | 0,01168856 |
| ENSMUSG00000026163 | Sphkap | −1,030633182 | 0,001174907 |
| ENSMUSG00000070304 | Scn2b | −1,038110113 | 0,009838673 |
| ENSMUSG00000042604 | Kcna4# | −1,228592873 | 0,000211934 |
| ENSMUSG00000029394 | Cdk2ap1 | −1,589028583 | 0,011081145 |

Genes in red: Fold DC vs. Fb is >2 x in *Sennett et al., 2015*; * indicates a DC signature gene.
Genes in blue: Fold Fb vs. DC is >2 x in *Sennett et al., 2015*; # indicates a Fibroblast signature gene.

DOI: https://doi.org/10.7554/eLife.36468.024

instead of FGF20 (*Figure 6—figure supplement 1*). Taken together, these data indicate that Fgf20 induces migration of embryonic dermal fibroblasts in vitro and that this effect is likely chemokinetic.

Next, we wanted to test the impact of Fgf20 in a more physiological setting. To this end, we introduced FGF20 locally via a bead on E13.5 dermis explants to mimic the in vivo situation in which Fgf20 is produced locally in the placode. First, we investigated the activity of FGF20 in this context by assessing its ability to upregulate the expression of *Spry4* and *Dusp6*, two known Fgf pathway feedback inhibitors also differentially expressed in our RNAseq data (*Figure 5*), using whole-mount RNA in situ hybridization. While we observed a robust induction of both genes after a 3 hr treatment, no consistent responses were detected after 8- and 16 hr treatments (*Figure 6H*), suggesting that the FGF20 protein loses its activity in intact dermal explants over this period of time. Vehicle-loaded control beads showed no induction of gene expression after any treatment period (*Figure 6H*). We further analyzed whether FGF20 bead incubation results in upregulation of *Sox2*, but observed no induction at any time point analyzed (*Figure 6H*). FGF9 beads led to a prominent induction of *Spry4* and *Dusp6* after all analyzed treatment periods (*Figure 6—figure supplement 1*). However, an overnight treatment also led to a robust increase in cell proliferation around the FGF9 bead (*Figure 6—figure supplement 1*). This response is in contrast to what we observed during dermal condensate formation in vivo (*Figure 3*) indicating that this experimental set-up may not represent a physiologically relevant model to study DC cell behavior and thus we concentrated on FGF20.

We next used the bead assay to test the impact of FGF20 on fibroblast cell behavior. To do this, we quantified the density of nuclei in E13.5 dermal explants cultured 3 hr with FGF20 or BSA vehicle control beads (*Figure 6D,E*). We observed a significant, 35% increase in cell density 15 μm around the FGF20 bead compared to the control bead (p=0.002), indicating that a localized Fgf20 source can induce aggregation of mesenchymal cells. We observed a similar increase in cell density upon treatment with FGF9 bead (p=0.033) (*Figure 6—figure supplement 1*). Nuclear shape analysis showed that cells in the immediate vicinity of the FGF20 bead displayed a significant decrease in nuclear sphericity compared to control bead (p<0.0001) (*Figure 6F,G*), similarly to DC cells in vivo (*Figure 1G,H*). We also tested whether cell cycle exit was induced upon local addition of FGF20. We could not detect *Cdkn1a* induction and no significant increase in Fucci-mKO+ cells were observed around the bead (*Figure 6—figure supplement 2*). Together this data suggests that Fgf20 can induce some of the cellular changes observed during DC morphogenesis. Finally, we analyzed whether ectopic expression of *Fgf20* could also lead to condensation of the mesenchyme in the *K14-Edar* model. We did not detect any significant difference in the cell density in the upper dermis between control, *K14-Edar*, and *K14-Edar;Fgf20⁻/⁻* genotypes (p>0.115) (*Figure 5—figure supplement 1*). However, given that *Cdkn1a* was expressed throughout the upper dermis in *K14-Edar* embryos, a lack of condensation in this model could be due to a lack of supply of cells available to condense.

## Inhibition of Fgf signaling impairs both recruitment and maintenance of DC cells ex vivo

The *Fgf20⁻/⁻* mouse model lacks all molecular and cellular signs of DC formation (*Huh et al., 2013*) (*Figure 1A,B*) and therefore offers limited tools to assess the effect of Fgf20 on cellular mechanisms governing DC formation. Therefore, we decided to inhibit Fgf signaling using SU5402 ex vivo which allows precise temporal manipulation of pathway activity followed by DC analysis in 3D. The same concentration of SU5402 that blocked the ability of Fgf20 to induce migration of E13.5 fibroblasts (*Figure 6*), also fully suppressed DC formation when applied to E13.5 skin cultured for 24 hr (*Figure 7—figure supplement 1*). Yet, it led to the expansion of *Fgf20^β-Gal* knock-in allele expression into a stripe-like pattern (*Figure 7—figure supplement 1*), as also observed in E14.5 *Fgf20⁻/⁻* embryos in vivo (*Huh et al., 2013*) confirming the applicability of this approach. SU5402 can also inhibit VEGFR and thus we used an inhibitor more specific to VEGFR, XL184, to test the effects of VEGFR-inhibition on DC formation. We added XL184 at an equivalent dose to inhibit VEGFR as 20 μM SU5402 (*Figure 7—figure supplement 2* and *Figure 7—figure supplement 2—source data 1*) as well as at five-times higher concentration and observed normal DC formation (*Figure 7—figure supplement 2D,E*). Finally, we confirmed that the absence of DCs in the SU5402-treated samples

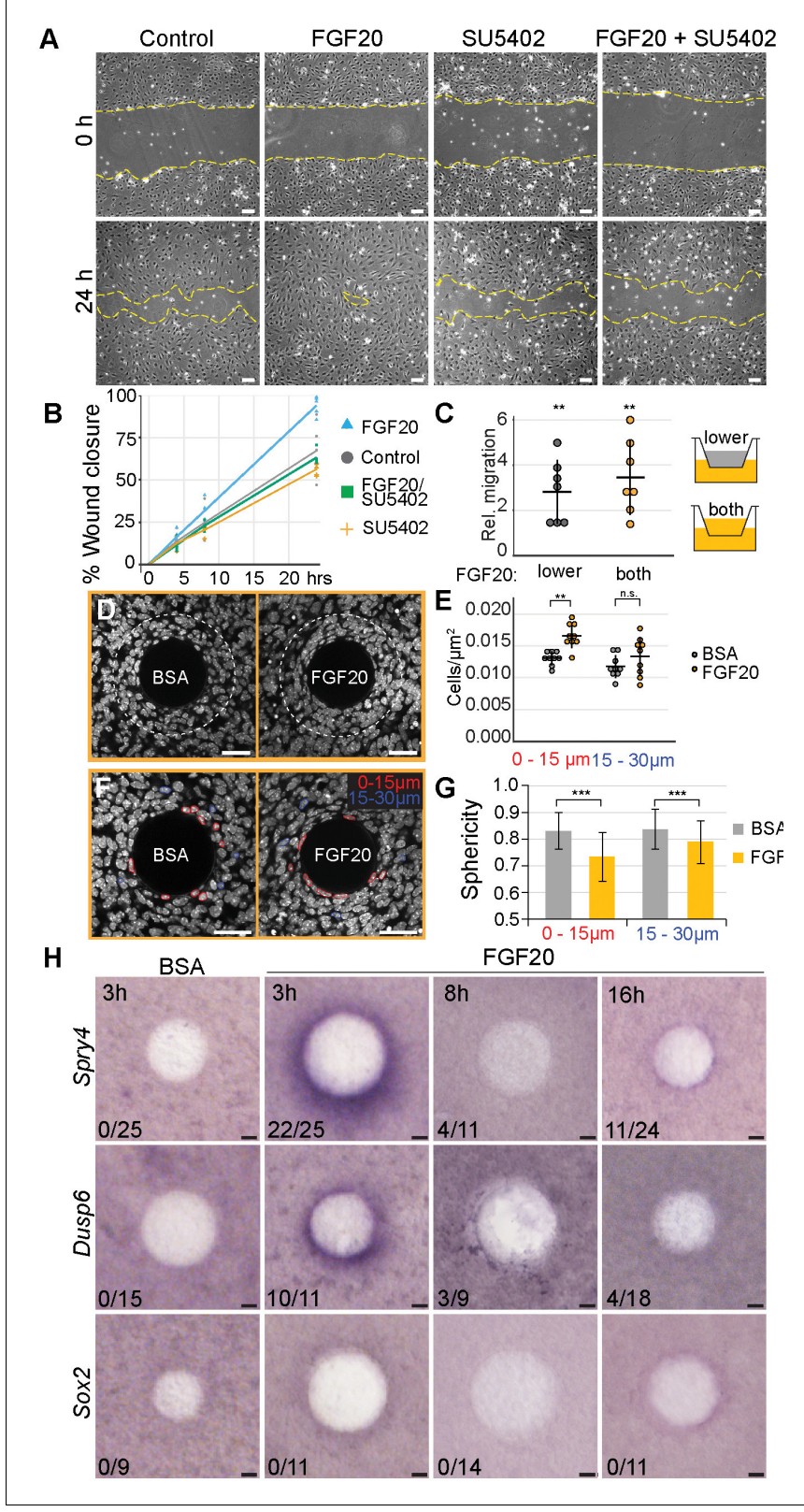

**Figure 6.** Fgf20 induces cell migration, condensation, and cell shape change ex vivo. (**A, C**) E13.5 cell cycle inhibited wild-type primary fibroblasts were utilized in scratch wound and transwell assays. (**A**) Phase-contrast images of scratch wounds (border denoted by dashed lines) at 0 hr (top) and 24 hr (bottom) post-wounding. The following treatments were added immediately prior to scratch induction: control (left) FGF20 (center left, 200 ng/

*Figure 6 continued on next page*

*Figure 6 continued*

ml), SU5402 (center right, 20 µM), or FGF20 +SU5402 (right). (**B**) Quantification of wound closure. At 24 hr, FGF20-treatment induced significantly faster wound healing relative to baseline control (closures 93.9 ± 5.73 and 67.3 ± 16.13% respectively, Student's t-test), and this effect was suppressed with SU5402 (closure 63.2 ± 4.22%). SU5402 alone had no effect on wound closure (all treatments n = 5 experiments each performed with freshly extracted primary dermal fibroblast cell population). (**C**) Transwell migration assay. Migration was significantly increased when Fgf20 (200 ng/ml) was added to lower or both upper and lower chambers (n = 7 experiments each performed with freshly extracted primary dermal fibroblast cell population, one-sample T-test). No statistical difference was observed between the two FGF20-treatments (Student's T-test). (**D, F**) Confocal images of E13.5 dermal explants cultured 3 hr with beads loaded with FGF20 or 0.1% BSA vehicle control and counterstained with Hoechst33342. Dashed line marks 30 µm radius from bead; nuclei within 15 µm (red) and 15–30 µm (blue) radii from beads. (**E**) Quantification of cell density from a single optical slice at mid-bead. FGF20 bead induced an increase in density within 15 µm radius from the bead relative to BSA control (paired Student's t-test,), but not between 15 and 30 µm radius (one-sample t-test, n = 9 beads). (**G**) Quantification of nuclear sphericity. Within 15 µm from the bead nuclear shapes of Fgf20 treated samples are significantly less spherical than control (data are from eight beads; $n_{BSA\ 0-15\mu m}$= 124; $n_{BSA\ 15-30\mu m}$ = 85; $n_{Fgf20\ 0-15\mu m}$=136; $n_{Fgf20\ 15-30\mu m}$ = 69 cells), significance was assessed with Mann-Whitney test. (**H**) Whole-mount RNA in situ hybridization of dermal samples cultured with FGF20 or 0.1% BSA control beads for 3, 8, or 16 hr. Induction of *Spry4* and *Dusp6*, but not *Sox2* expression (purple) was observed around the bead at 3 hr. n indicates induction/total samples, induction was tested in two independent experiments with skin samples derived from ≥2 different litters. n.s., not significant *, p≤0.05; **, p≤0.01; ***, p≤0.001. Error bars represent SD. Scale bars: A = 100 µm; D, F, and H = 30 µm. See also *Figure 6—source data* and *Figure 6—figure supplement 1*.

DOI: https://doi.org/10.7554/eLife.36468.025

The following source data and figure supplements are available for figure 6:

**Source data 1.** Values used to quantify FGF20-induced cellular changes.
DOI: https://doi.org/10.7554/eLife.36468.030

**Figure supplement 1.** Dermal fibroblasts migrate in vitro and condense ex vivo in response to FGF9-treatment.
DOI: https://doi.org/10.7554/eLife.36468.026

**Figure supplement 1—source data 1.** Values used to quantify FGF9-induced cellular changes.
DOI: https://doi.org/10.7554/eLife.36468.027

**Figure supplement 2.** FGF20 does not induce *Cdkn1a* expression nor Fucci^mKO+-cells.
DOI: https://doi.org/10.7554/eLife.36468.028

**Figure supplement 2—source data 1.** Values used to quantify FGF20 or FGF9 induced *Fucci-mKO* expression.
DOI: https://doi.org/10.7554/eLife.36468.029

was due to inhibition of FGFR-signaling by using BGJ398, an inhibitor more specific to FGFR (*Figure 7—figure supplement 2*). An equivalent dose to inhibit FGFR as 20 µM SU5402 and a 2.5-times higher dose blocked formation of the DCs in the skins and altered the *Fgf20$^{\beta-Gal}$* knock-in allele expression similar to the *Fgf20$^{-/-}$* animals (*Figure 7—figure supplement 2*), confirming the effects of the SU5402 to be due to FGFR-inhibition.

Next, we applied SU5402 slightly later, when DC formation had initiated. Skin explants were divided into two halves: one was used as the control and the other one treated with SU5402 (*Figure 7A* and *Figure 7—figure supplement 1*). At this stage, the inhibitor did not result in absence of dermal condensates, but the number of Sox2+ cells was significantly lower in the SU5402-treated samples compared to the controls (p=0.0061) (*Figure 7B*), indicating that Fgf signaling is necessary for further addition of Sox2+ cells. Analysis of the average distances of DC cells to their nearest neighbor, however, showed no difference between control and SU5402 treated samples (p=0.7319) (*Figure 7C*) suggesting that inhibition of Fgf signaling does not affect the density of the existing DC. To assess whether Fgf20 could also play a role in the maintenance of DC cells, we compared dermal condensates at the start (T0) of experiment with condensates after 12 hr SU5402 treatment (*Figure 7D* and *Figure 7—figure supplement 1*). A significant reduction in the number of Sox2+ DC cells was observed (p=0.0036) (*Figure 7E*). Again, Sox2+ DC cell density was not affected by SU5402 treatment (p=0.3010) (*Figure 7F*). Taken together, these results highlight the role of Fgf signaling both in recruitment and maintenance of DC cells.

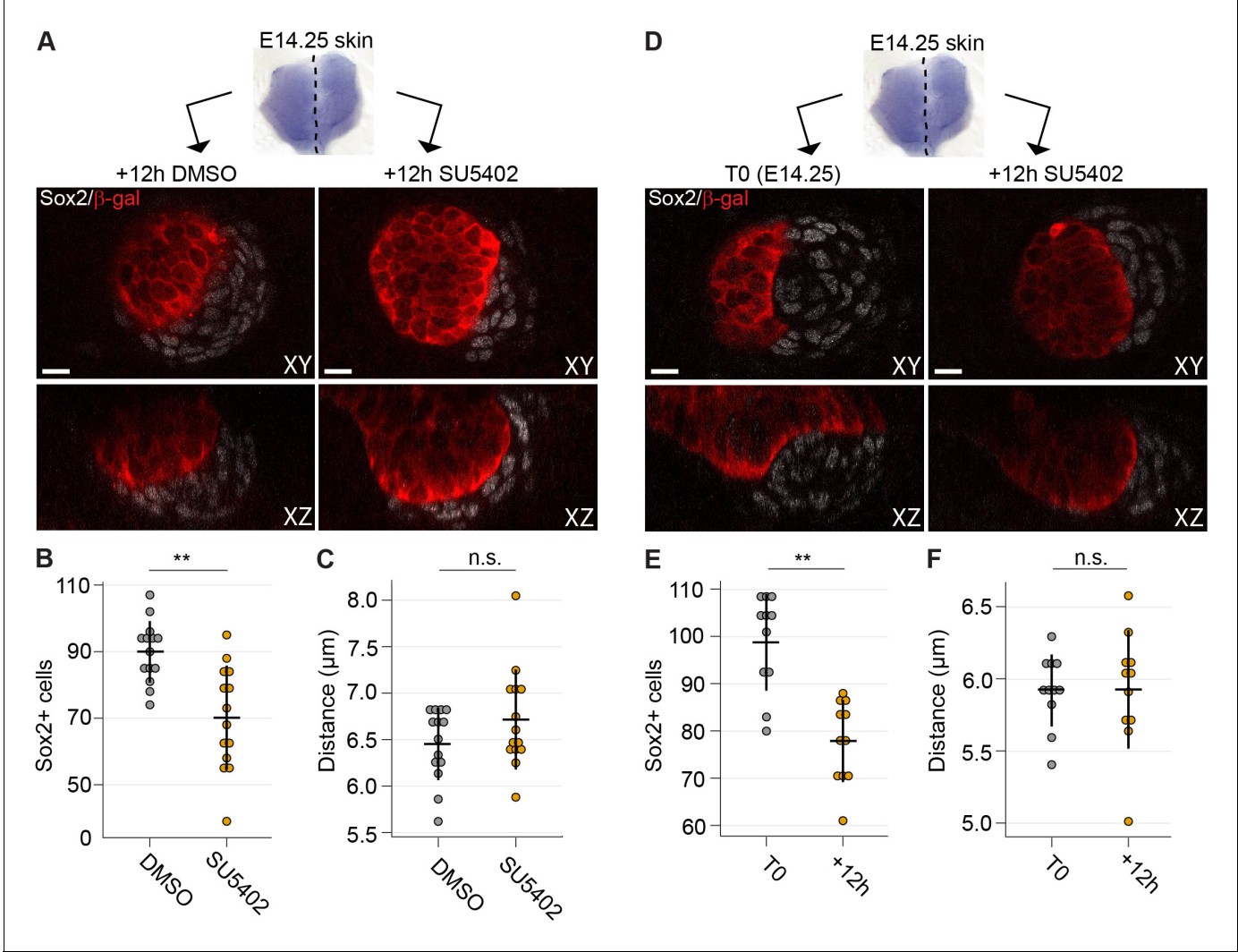

**Figure 7.** Inhibition of Fgf signaling blocks morphogenesis of dermal condensate ex vivo. (A) Confocal optical sections of paired E14.25 *Fgf20*⁺/⁻ skins explants cultured for 12 hr with SU5402 (20 μM) or DMSO vehicle control. Samples were stained for β-Gal (red) and Sox2 (white). (B) Quantification of Sox2+ DC cell numbers in control and SU5402-treated samples relative to DMSO control (n = 14 DCs each from six skins). (C) Quantification of the distance of Sox2+ DC cells to their nearest neighbor in control and SU5402-treated samples (n = 14 DCs each from six skins). (D) Confocal optical sections of E14.25 paired skin explants either fixed at T0 or cultured 12 hr with SU5402 (20 μM). Samples were stained for β-gal (red) and Sox2 (white). (E, F) Quantification of Sox2+ DC cell numbers and distance to neighbor (n = 11 DCs from four skins). Significance was assessed with Student's T-test. n.s., not significant; **, p≤0.01. Error bars represent SD. See also *Figure 7*—source data and *Figure 7—figure supplement 1*.

DOI: https://doi.org/10.7554/eLife.36468.031
The following source data and figure supplements are available for figure 7:

**Source data 1.** Values used to quantify DC morphogenesis in the presence of Fgfr inhibitor.
DOI: https://doi.org/10.7554/eLife.36468.035
**Figure supplement 1.** Inhibition of Fgf signaling at different developmental stages ex vivo impairs DC formation and maintenance.
DOI: https://doi.org/10.7554/eLife.36468.032
**Figure supplement 2.** Inhibition of VEGFR signaling with XL184 and FGFR signaling with BGJ398 in skin explants.
DOI: https://doi.org/10.7554/eLife.36468.033
**Figure supplement 2—source data 1.** Inhibitors used to test FGFR signaling in DC induction.
DOI: https://doi.org/10.7554/eLife.36468.034

## Discussion

Mesenchymal (dermal) condensation is a common phenomenon occurring during organogenesis including most ectodermal appendages such as tooth, mammary gland, feather, and hair follicle

(*Biggs and Mikkola, 2014*). Despite the critical role of dermal condensation for organ development, the underlying mechanisms have remained elusive (*Hall and Miyake, 2000*; *Widelitz and Chuong, 1999*; *Newman and Bhat, 2007*; *da Rocha-Azevedo and Grinnell, 2013*). Here, we have used 3D and 4D imaging to study DC morphogenesis in murine pelage primary hair follicles. The data presented here support two key conclusions. First, our findings show that early morphogenesis of the HF DC is characterized by cell shape changes, exit of the cell cycle, and directed migration. Second, our data indicate that Fgf20 regulates these processes, further corroborating the critical role of Fgf20 in HF dermal condensation. Our results reveal ~2-fold higher cell density in the DC compared to the interfollicular dermis. Ablation of Fgf20 results in the absence of all the DC markers analyzed thus far (*Huh et al., 2013*), and here, we show by 3D analysis that the cells below the primary placode show no evidence of condensation. These data confirm that DC marker expression and fibroblast condensation are closely associated, although whether DC fate and condensation can be uncoupled remains unclear. Furthermore, this finding supports our previous conclusions (*Huh et al., 2013*) that hair follicle patterning is governed by and first arises in the epithelium rather than the mesenchyme.

## DC does not form from a pre-existing pool of Sox2+ cells

The origin of the DC/DP population is poorly understood. Tissue recombination and mouse mutant studies (reviewed in [*Biggs and Mikkola, 2014*; *Morgan, 2014*]) together with the ex vivo Fgf inhibitor experiments (this study) show that early DC fate is plastic and reversible. Yet, these findings do not preclude the existence of a predetermined pool of DC cells. Previous studies have shown that DPs of the dorsum largely derive from the early fibroblast precursor population marked by expression of *Delta-like homologue 1* (*Dlk1*) (*Driskell et al., 2013*), and are polyclonal in origin (*Collins et al., 2012*). Dlk1 lineage tracing from E12.5 (when all dermal fibroblasts are Dlk1+), but not after E16.5, reveal a contribution to post-natal DP (*Driskell et al., 2013*). Lineage tracing of neural crest cells with *Wnt1-Cre* or *Ht-PA-Cre* reveals that the whisker DP (along with non-DC head/facial fibroblasts) is neural crest-derived, but that neural crest cells or their progeny are rare in back skin DP (*Fernandes et al., 2004*; *Wong et al., 2006*). Our short-term lineage-tracing experiments show that Sox2 expression is de novo acquired in DC cells, confirming that the dorsal DCs do not arise from a pre-existing pool of Sox2+ cells, for example the neural crest-derived Schwann cell lineage of the skin (*Adameyko et al., 2012*; *Sennett et al., 2015*). Studies in an adult model of ectopic hair follicles via forced epithelial β-catenin also argue against a pre-specified DC/DP subpopulation of dermal fibroblasts (*Collins et al., 2012*). Hence, all available data suggest that the unique attributes of DC/DP cells do not reflect a distinct fibroblast lineage but are induced by placode-derived signals.

## Cell shape change and cell cycle exit are early hallmarks of HF dermal condensation

We show in 3D that HF DCs exhibit a less spherical shape in vivo, and that this shape change is an early event during DC morphogenesis. Previous studies in other organs (bone, tooth, feather) have also revealed an altered cell shape in condensed mesenchyme when compared to the adjacent non-condensed tissue (*Thorogood and Hinchliffe, 1975*; *Ray and Chapman, 2015*; *Mammoto et al., 2011*; *Wessells, 1965*). However, these analyses were conducted in 2D and thus their similarity or difference with the HF DC is not apparent. What is of note is that the cells within condensed mesenchyme display a distinct morphology. Indeed, a critical role for actomyosin contractility has been proposed to drive cytoskeletal rearrangements, and that the resulting cell shape changes are required for mesenchymal condensation (*Ray and Chapman, 2015*). Here, we show that in the hair follicle DC the intensity of F-actin is increased in the DC compared to the surrounding mesenchyme, likely playing a critical role in maintaining cell shape. We show that in hair follicle DC, the cell shape changes depend on and can be induced by Fgf20. The utility of this cell shape change along with cell compaction could be manifold, including increased cell-cell contacts to foster cell-cell communication and further to maintain the structure of the DC. Transcriptomic analysis has indicated the expression of cell-cell adhesion factor *R-cadherin* and *cadherin11* (encoded by *Cdh4* and *Cdh11*, respectively) in the DC (*Sennett et al., 2015*) and cadherin-based junctions were detected between

DP cells at E17 (*Nanba et al., 2003*), but their functional importance for condensation remains to be tested.

During DC morphogenesis, Sox2+ cells were found to exhibit a rapid exit from the cell cycle and maintain the $G_0/G_1$ status through E15.5, a finding in line with a previous study showing that morphologically distinct DC cells fail to incorporate $H^3$-thymidine (*Wessells and Roessner, 1965*). Even as the size of the DC is increased by genetic means (*K14-Eda*), the DC cells remain quiescent supporting the idea that the increase in DC cell number is not a result of proliferation. Further, mitotic inactivity is a prominent feature of the mature HF DP (*Pierard and de la Brassinne, 1975*). During the growth (anagen) and rest (telogen) phases of the hair cycle, however, the DP dynamically increases and decreases in cell number. Yet, the increase in DP cell number upon reentering anagen phase does not arise via proliferation but instead DP cells are recruited from the proximal dermal sheath cells (*Tobin et al., 2003*; *McElwee et al., 2003*; *Chi et al., 2010*). Furthermore, hair reconstitution experiments show that mitotically inhibited cells are fully competent to generate the DP (*Collins et al., 2012*) indicating that proliferation is not necessary for DC/DP formation.

We have previously shown that expression of *Cdkn1a* is an early marker of DCs (*Huh et al., 2013*), and recent transcriptome profiling study showed that *Cdkn1c* (a.k.a. *p57*), another cyclin dependent kinase inhibitor, is also expressed at high levels in the DC (*Sennett et al., 2015*). Interestingly, Fgf20 target transcriptome analysis displayed upregulation of cell-cycle-related genes, such as *Cdkn1a* and *Bcl6*, which suggests that Fgf signaling could provide the cue for the $G_0$-arrest observed in the DC fibroblasts. Despite this, FGF20 was unable to induce cell cycle arrest in the dermis in a short-term experiment when supplied in a localized manner. However, *Cdkn1a* was ectopically expressed in our model of *Fgf20* overexpression (*K14-Edar*) in an *Fgf20*-dependent manner. Similarly, Fgf signaling induces *Cdkn1a*-mediated cell cycle arrest in chondrocytes (*Aikawa et al., 2001*) and further, ectopic Fgf20 induces growth arrest of rat chondrosarcoma cells in vitro (*Buchtova et al., 2015*). *Bcl6* is also a DC signature gene (*Sennett et al., 2015*) and although its function is context-dependent, it has been shown to suppress proliferation of many primary cells including fibroblasts (*Ranuncolo et al., 2008*).

## Directed cell migration drives dermal condensate morphogenesis

Our confocal time-lapse imaging of developing HF revealed that dermal fibroblasts initially outside of the future condensate migrate toward the placode, indicating directed migration as a mechanism of DC formation. A recent study tracking movements of Wnt-responsive cells using *TCF/Lef::H2B-GFP* reporter (*Glover et al., 2017*) is consistent with our findings. In vitro scratch wound assay revealed that Fgf20 enhances cell motility. Although transwell assays did not support a chemotactic role for Fgf20, we showed that local delivery of FGF20 via beads on dermal explants induces an increase in cell density, suggesting that Fgf20 induces directed migration and in a tissue context, dermal condensation. Further, we find that inhibition of Fgfr signaling ex vivo blocks accumulation of Sox2+ fibroblasts in the DC while Fgf20 does not appear to directly regulate *Sox2* expression. These data support the conclusion that Fgf20 signaling regulates directed movement of dermal fibroblasts. An additional factor that we did not test but may play a role in DC morphogenesis is the contribution of differential ECM composition surrounding the hair follicle (*Kaplan and Holbrook, 1994*; *Pflieger et al., 2006*). It is possible that Fgf20 induces migration of fibroblasts and simultaneously, differential ECM composition around the hair follicle holds the DC cells in place.

Previous studies have shown that Fgf signaling induces migration, both chemotactic and chemokinetic, in several different developmental contexts (*Mammoto et al., 2011*; *Delfini et al., 2005*; *Attia et al., 2012*). Indeed, a general role for Fgf signaling in regulating dermal condensation via cell migration is an appealing idea. Our ex vivo manipulation experiments showed that inhibition of Fgfr signaling suppresses accumulation of new Sox2+ cells, but also compromises maintenance of nascent DC. The latter finding is suggestive for a role also in DC maintenance. In odontogenic mesenchyme, attractive Fgf8 and repellent Sema3f are thought to act in concert to induce migration-driven cell compaction (*Mammoto et al., 2011*). Involvement of Fgf signaling in mammary mesenchyme condensation is an intriguing hypothesis, but has not been examined. *Fgf20* is expressed in the epithelium of the mammary buds during the period of mesenchymal condensation (*Elo et al., 2017*) yet it seems to be dispensable for this process, but other epithelially expressed Fgfs may compensate for the loss of Fgf20. In contrast, Fgf20 is necessary for feather development (*Houghton et al., 2007*; *Wells et al., 2012*). Although the exact function of Fgf20 in feather

morphogenesis has not been studied, the ability of exogenous Fgfs to induce dermal cell aggregation ex vivo (*Lin et al., 2009*; *Song et al., 2004*) and to elicit feather formation in Fgf20-deficient skin explants (*Song and Sawyer, 1996*) argue for a conserved role for Fgf20 in DC induction.

## Fgf20 signaling induces partial DC fate

Although the transcriptional profile of the DC has been described (*Sennett et al., 2015*), the molecular regulation of DC fate acquisition is not well understood. Fgf20 is necessary for DC marker expression (*Huh et al., 2013*) and here we show that Fgf20 is also necessary for dermal cell condensation. However, our RNAseq profiling of genes induced in the dermis upon short Fgf20 treatment revealed only a few DC signature genes (*Sennett et al., 2015*), and for example Sox2, was not upregulated by Fgf20 ex vivo, nor in our in vivo over-expression model of ectopic Fgf20 signaling. These data suggest that Fgf20 alone is sufficient to induce only a subset of DC markers. It is possible that other cues, molecular or mechanical, together with Fgf20 determine DC fate, and studies in dental and cartilage mesenchyme show that cellular condensation contributes to fate acquisition (*Mammoto et al., 2015*; *Mammoto et al., 2011*; *Ray and Chapman, 2015*). Alternatively, it is possible that Fgf20 functions mainly to regulate cell behaviors rather than fate.

In conclusion, we show here that cell shape change, cell cycle exit and directed migration define HF DC formation. While altered cell shape and directed mesenchymal cell movement may be a shared characteristic, cell cycle exit appears to be a hair follicle specific feature, as cells within the condensing mammary (*Lee et al., 2011*) and tooth mesenchyme exhibit proliferation (*Mammoto et al., 2011*) at the same rate as the surrounding non-condensed mesenchyme. However, the function of DC cell cycle exit in hair follicle morphogenesis remains unknown. It appears not to be sufficient to induce DC fate, but it remains open whether it is necessary. To date, culturing methods for maintaining the hair-follicle inductive capacity of DC/DP remain elusive, and after a few passages, DP populations completely lose their inductive abilities. Although speculative, the obligate proliferation under in vitro culture may compromise DC fate maintenance. The challenge in future therapeutic efforts to generate hair inductive fibroblasts is to uncover culture conditions that induce DC cell fate de novo.

# Materials and methods

**Key resources table**

| Reagent type (species) or resource | Designation | Source or reference | Identifiers | Additional information |
|---|---|---|---|---|
| Strain, strain background (*Mus musculus*, C57/Bl6) | $Fgf20^{+/-}$ | PMID: 23431057 | RRID:MGI:5425887 | |
| Strain, strain background (*M. musculus*, C57/Bl6) | $Fgf20^{-/-}$ | PMID: 23431057 | RRID:MGI:5425887 | |
| Strain, strain background (*M. musculus*, C57/Bl6) | K14-Eda | PMID: 12812793 | | |
| Strain, strain background (*M. musculus*, C57/Bl6) | $R26R^{tdTomato}$ | Jackson Laboratory | Stock 007914, RRID:IMSR_JAX:007914 | |
| Strain, strain background (*M. musculus*, C57/Bl6) | $R26R^{mT/mG}$ | Jackson Laboratory | Stock 007576, RRID:IMSR_JAX:007576 | |
| Strain, strain background (*M. musculus*, C57/Bl6) | $Sox2^{creERT}$ | Jackson Laboratory | Stock 017593, RRID:IMSR_JAX:017593 | |
| Strain, strain background (*M. musculus*, mixed) | Fucci | PMID: 18267078 | RRID:IMSR_RBRC02892 | |

*Continued on next page*

*Continued*

| Reagent type (species) or resource | Designation | Source or reference | Identifiers | Additional information |
|---|---|---|---|---|
| Strain, strain background (*M. musculus*, C57/Bl6) | *R26Fucci2aR* | EMMA | EM:08395, RRID:IMSR_EM:08395 | |
| Strain, strain background (*M. musculus*, C57/Bl6) | *Sox2-GFP* | PMID: 12923297 | | |
| Strain, strain background (*M. musculus*, C57/Bl6) | *K14-Edar* | PMID: 15366021 | | |
| Antibody | Beta-galactosidase rabbit | MP Biomedicals | 0855976, RRID:AB_2334934 | 1:1500 |
| Antibody | Beta-galactosidase chicken | Abcam | ab9361, RRID:AB_307210 | 1:1500 |
| Antibody | EpCAM rat monoclonal | BD Pharmingen | 552370, RRID:AB_394370 | 1:500 |
| Antibody | Krt14 rabbit monoclonal | Thermo Fisher Scientific | RB-9020-P, RRID:AB_149790 | 1:500 |
| Antibody | Sox2 goat polyclonal | Santa Cruz | SC-17320, RRID:AB_2286684 | 1:500 sections, '1:200 wholemount |
| Antibody | Sox2 rabbit polyclonal | Stemgent | 09–0024, RRID:AB_2195775 | 1:300 |
| Antibody | *Cdkn1a* rabbit monoclonal | Abcam | ab188224, RRID:AB_2734729 | 1:1000 |
| Sequence-based reagent | *Dusp6* RNA probe | PMID: 11960712 | | |
| Sequence-based reagent | *Cdkn1a* RNA probe | PMID: 9486790 | | |
| Sequence-based reagent | *Spry4* RNA probe | PMID: 11731251 | | |
| Sequence-based reagent | *Sox2* RNA probe | PMID: 15240551 | | |
| Sequence-based reagent | *Edar* RNA probe | PMID: 11203701 | | |
| Sequence-based reagent | *Hprt* probe | BioRad | qMmuCEP0054164 | |
| Sequence-based reagent | *Dusp6* probe | BioRad | qMmuCIP0029423 | |
| Sequence-based reagent | *Eef1* probe | BioRad | qMmuCEP0057829 | |
| Sequence-based reagent | *Spred1* probe | BioRad | qMmuCEP0055028 | |
| Sequence-based reagent | *Gapdh* probe | BioRad | qMmuCEP0039581 | |
| Sequence-based reagent | *Etv5* probe | BioRad | qMmuCIP0034710 | |
| Sequence-based reagent | *Spry4* probe | BioRad | qMmuCEP0054507 | |
| Sequence-based reagent | *Bcl6* qPCR primers | this study | template NM_001348026.1 | F: CGCGAACCTTGATCTCCAGT, R: CAGGGACCTGTTCACGAGAT |
| Sequence-based reagent | *Hprt* qPCR primers | this study | template NM_013556.2 | F: CAGTCCCAGCGTCGTGATTA, R: TCGAGCAAGTCTTTCAGTCCT |
| Peptide, recombinant protein | FGF20 human recombinant protein | PeproTech | 100–41 | |

*Continued on next page*

*Continued*

| Reagent type (species) or resource | Designation | Source or reference | Identifiers | Additional information |
|---|---|---|---|---|
| Peptide, recombinant protein | FGF9 human recombinant protein | R and D Systems | 273-F9 | |
| Commercial assay or kit | Rneasy plus micro kit | Qiagen | ID: 74004 | |
| Commercial assay or kit | Quantitect reverse transcription kit | Qiagen | ID: 205311 | |
| Commercial assay or kit | TruSeq Stranded Total RNA Library Prep Kit with Ribo-Zero Mouse | illumina | RS-122–2202 | |
| Chemical compound, drug | SU5402 | Calbiochem | 572630 | |
| Chemical compound, drug | BGJ398 | Selleckchem.com | S2183 | |
| Chemical compound, drug | XL184 | Selleckchem.com | S4001 | |
| Software, algorithm | AfterQc | PMID: 28361673 | RRID:SCR_016390 | |
| Software, algorithm | SortMeRNA | PMID: 23071270 | RRID:SCR_014402 | |
| Software, algorithm | STAR | PMID: 23104886 | RRID:SCR_015899 | |
| Software, algorithm | Htseq-count | PMID: 25260700 | RRID:SCR_011867 | |
| Software, algorithm | DEseq2 | PMID: 25516281 | RRID:SCR_015687 | |

### Ethics statement

All mouse studies were approved and carried out in accordance with the guidelines of the Finnish national animal experimentation board.

### Mouse lines

The following mice were maintained on C57Bl/6 background. *Fgf20*$^{β-Gal}$ mice harbor an Fgf20-β-Galactosidase knock-in allele (*Huh et al., 2013*); *Sox2*$^{CreER}$ was obtained from Jackson Laboratory (Stock 017593); *R26R*$^{tdTomato}$ was obtained from Jackson Laboratory (Stock 007914); *R26R*$^{mT/mG}$ was obtained from Jackson Laboratory (Stock 007576). *R26Fucci2aR* mice (*Mort et al., 2014*) were obtained from the European Mouse Mutant Archive (EM:08395) and upon derivation mated with ubiquitous cre line, *Pgk1-cre* (Jackson Laboratory; Stock 020811) in order to obtain mice which heritably and constitutively express the Fucci2a construct in every cell. *Sox2-EGFP*, *K14-Eda*, and *K14-Edar* have been described (*Mustonen et al., 2003*; *Pispa et al., 2004*, *D'Amour and Gage, 2003*). *Fucci* cell cycle indicator mice (*Sakaue-Sawano et al., 2008*) were maintained on a mixed background.

Mice were kept in 12 hr light-dark cycles and food and water were available *ad libitum*. To label *Sox2*-expressing cells, pregnant wild-type dams mated with *Sox2*$^{CreER/wt}$; *R26R*$^{tdTomato/tdTomato}$ males were given one intraperitoneal injection of 3 mg tamoxifen (Sigma-Aldrich, Saint Louis, MO) dissolved in corn oil (Sigma-Aldrich) at 12pm on the indicated day of pregnancy (appearance of a vaginal plug was taken as embryonic (E) 0). All embryos used in the study were staged according to limb and other external morphological criteria.

### Transmission electron microscopy

Samples were fixed in 2.5% glutaraldehyde at room temperature for 2 hr, washed in 0.1 M NaPO$_4$, and subsequently fixed in 2% PFA in 0.1 M NaPO$_4$. The samples were then dehydrated through a graded series of ethanol and acetone before embedding in Epon. Ultra-thin sections were generated. Images were acquired with Jeol JEM-1400 electron microscope (Jeol Ltd., Tokyo, Japan).

## In situ hybridization

For whole-mount RNA in situ hybridization, cultured dermal explants were fixed to their filter with cold methanol for 2 min and then fixed in 4% PFA in PBS overnight at 4°C, washed with PBS and then dehydrated in a series of methanol. The hybridization was performed using InSitu Pro robot (Intavis AG, Cologne, Germany) as described before (*Fliniaux et al., 2008*; *Huh et al., 2013*). Briefly, the samples were rehydrated in a methanol series, treated with 10 µg/ml Proteinase K (Roche, Mannheim, Germany) for 5 min and post fixed with 4% PFA. The hybridization was performed with digoxigenin-labeled antisense RNA probes: *Dusp6* (*Dickinson et al., 2002*), *Cdkn1a* (*Jernvall et al., 1998*), *Spry4* (*Zhang et al., 2001*), and *Sox2* (*Ferri et al., 2004*), 1 µg/ml in hybridization buffer at 65°C for 14 hr. After hybridization, the excess probe was removed in stringent washes, samples were blocked, the probe was detected with an alkaline phosphatase conjugated anti-digoxigenin antibody (Roche, Mannheim, Germany), and a subsequent reaction with precipitating alkaline phosphatase substrate BM purple (Roche, Mannheim, Germany). Samples were fixed with 4% PFA and imaged using Lumar.V12 stereomicroscope with 1.2x objective and AxioCam ICc camera (Zeiss, Oberkochen, Germany).

For radioactive section in situ hybridization, E16.5 embryos were collected, fixed in 4% PFA in PBS and processed into paraffin blocks using standard protocols and cut into 5 µm sagittal sections. Radioactive in situ hybridization with $^{35}$S-UTP-labeled probes: *Edar* (*Laurikkala et al., 2001*), *Sox2* (*Ferri et al., 2004*), and *Cdkn1a* (*Jernvall et al., 1998*), was performed as previously described (*Huh et al., 2013*). Sections were imaged using Axio Imager M.2 widefield microscope equipped with Plan-Neofluar 20x/0.5 objective and AxioCam HRc camera (Zeiss) using bright and dark field microscopy. The dark field images were inverted, thresholded linearly and superimposed on the bright field images using Adobe Photoshop software (Adobe, San Jose, CA).

## X-gal staining

*Fgf20*$^{β-Gal/+}$ embryos were pre-fixed for 2 hr in 2% PFA, 0.2% glutaraldehyde (Sigma-Aldrich, St. Louis, MO) in PBS, 4°C and then rinsed with PBS and washed 3 × 15 min with PBS, 2 mM MgCl$_2$ (Merck, Darmstadt, Germany), 0.02% NP-40 (Sigma-Aldrich), 4°C. Subsequently, the samples were stained for 10 hr at RT, in dark with 1 mg/ml X-Gal (Thermo Fischer Scientific, Vilnius, Lithuania), 5 mM K$_3$Fe(CN)$_6$ (Merck, Darmstadt, Germany), 5 mM K$_4$Fe(CN)$_6$ (Merck, Darmstadt, Germany), 2 mM MgCl$_2$, 0.1 % NP-40 (Calbiochem, San Diego, CA), 0.2% Na-deoxycholate (Sigma-Aldrich, St. Louis, MO) in PBS. The embryos were washed 3 × 10 min with PBS and fixed with 4% PFA in PBS at RT. After imaging, the samples were processed for paraffin blocks using standard protocols and sectioned into 5 µm sagittal sections. The sections were deparaffinized and counter stained with Nuclear Fast Red (Sigma-Aldrich, Steinheim, Germany) for 5 min, dehydrated and mounted. The sections were imaged using Axio Imager M.2 wide field microscope equipped with Plan-Neofluar 10X/ 0.3 objective and AxioCam HRc camera (Zeiss).

## In vitro fibroblast scratch wound healing assay, and transwell migration assay

Primary dermal fibroblasts were extracted from E13.5 wild-type NMRI mouse embryos. Briefly, the back skin was dissected and treated with 2.5 mg/ml Pancreatin (Sigma-Aldrich), 22.5 mg/ml Trypsin (Difco, Sparks, MD) in Tyrode's solution for 8 min at RT, followed by an incubation with 10% FBS in DMEM (Gibco by Life Technologies, Paisley, UK) for 1 hr at RT. The tissues were manually separated; epidermis was discarded and the mesenchyme was gently dissociated by pipetting in 0.2% FBS in DMEM. For scratch wound healing assay, the fibroblasts were seeded on fibronectin-coated plates (1 µg/cm$^2$, R and D Systems, Minneapolis, MN) at the density of 125,000 /cm$^2$ and cultured for 16 hr in 0.2% FBS, 1% penicillin-streptomycin (Life Technologies, Eugene, OR) in DMEM. Cell cycle inhibition was achieved by a 2 hr treatment with 5 ng/ml aphidicolin (Sigma-Aldrich, Jerusalem, Israel). Scratches were induced with a pipette tip and the cells were treated with human recombinant FGF20 (200 ng/ml, Peprotech, Rocky Hill, NJ), human recombinant FGF9 (200 ng/ml, R and D Systems), SU5402 (20 µM, Calbiochem, Darmstadt, Germany) or BSA (0.1%). 4 µg/ml heparin (Sigma-Aldrich, St. Louis, MO) was added with the FGF proteins. Conditions were carried out in duplicate in five independent experiments. Wounds were imaged at 0, 4, 8, and 24 hr using Leica DM IRB phase contrast microscope (Leica Microsystems), and ImageJ (http://rsbweb.nih.gov/ij/) was used to

manually measure the open area at the indicated time points in two locations. Proportion of closure was determined as the ratio of open area at each time point compared to 0 hr. For transwell migration assay, 50,000 freshly isolated cells were seeded in 300 µl of DMEM, 1% FBS in cell culture inserts for 24-well plates (Millicell, Basel, Switzerland) and cultured overnight. The cells were then treated with 200 ng/ml FGF20, FGF9 or 0.1% BSA vehicle control (baseline) in DMEM containing 1% FBS and 5 ng/ml aphidicolin; conditions were carried out in duplicate in six independent experiments. The proteins were introduced either in the receiving compartment or both receiving and seeding compartments and cells were allowed to migrate for 8 hr. The seeding compartment side of the membrane was mechanically cleared of cells; the remaining cells were fixed with methanol 5 min, and the membrane was stained with 0.1% Crystal violet (Sigma-Aldrich, St. Louis, MO), 70% ethanol in RO-water. Absolute cell number was counted at five positions on each membrane. Relative migration was determined as the average number of cells in the duplicate inserts divided by average number of cells in the duplicate inserts.

## Hanging drop culture, RNA sequencing, and qRT-PCR

E13.5 skins were dissected from $Fgf20^{-/-}$ embryos and enzymatically separated using 2.5 U/ml Dispase II (Roche by Godo Shusei, Tokyo, Japan) in 4°C and 30 min resting in culture medium, followed by mechanical separation. Each dermis was cut into two halves along the midline and one half was incubated in FGF20 (1 µg/ml) and the other half was incubated in 0.1% BSA vehicle control in 0.1% FBS, heparin (2 µg/ml), 1% penicillin-streptomycin in DMEM. The tissues were incubated in hanging drops of the indicated media for 3 hr at 37°C, 5% $CO_2$. For RNAseq, five biological replicates were used. The samples were stored in RNAlater (Qiagen GmbH, Hilden, Germany). RNA was extracted using RNeasy Plus micro kit (Qiagen GmbH, Hilden, Germany), according to manufacturer's instructions. RNA quality was assessed with 2100 Bioanalyzer (Agilent, Santa Clara, CA) and RIN values averaged 9.6. The cDNA libraries were prepared with TruSeq Stranded Total RNA with RiboZero (Illumina, San Diego, CA), and sequenced with NextSeq500 (Illumina, San Diego, CA). The Illumina-seq reads produced around 35 million reads for each sample. The quality of each sample was assessed with FastQC and processed with AfterQC (*Chen et al., 2017*). The ribosomal RNAs were filtered out with SortMeRNA (*Kopylova et al., 2012*). Thereafter the reads that passed the QC threshold were mapped to mouse genome (GRCm38/mm10/Ensembl release 79 - March 2015) using STAR mapping tool (*Dobin et al., 2013*) and on average 81% reads uniquely mapped. The expression of genes and differentially expressed genes (adjusted p value<0.05) were measured by HTseq-count (*Anders et al., 2015*) and DEseq2 (*Love et al., 2014*). Access to the data set is found at https://www.ncbi.nlm.nih.gov/geo/query/acc.cgi?acc=GSE110459.

For qRT-PCR, 1 µg of RNA was used for cDNA synthesis with the QuantiTect Reverse Transcription Kit (Qiagen), according to manufacturer's instructions. cDNA was diluted to 10 ng/µl and 40 ng was used for one qPCR reaction. Probe-based multiplex RT-qPCR reactions were prepared in 1X iTaq Universal Probes Supermix (Bio-Rad, Hercules, CA) using a validated combination of gene-specific PrimePCR Probe Assays (Bio-Rad, Hercules, CA) in a 20 µl volume. The probe combinations are listed in Supplementary file 1. Reactions from all samples were run in triplicate wells with the CFX96 Real-Time System (Bio-Rad, Hercules, CA) using the following cycling protocol: Initial denaturation 3 min at 95°C, followed by 45 cycles of denaturation 10 s at 95°C and annealing/extension 50 s at 60°C. Fold changes were calculated with the ΔΔCT method (*Livak and Schmittgen, 2001*). These were normalized to reference genes as follows: *Etv5* and *Spry4* were normalized to *Gapdh*, *Cdkn1a* and *Dusp6* were normalized to *Hprt,* and *Spred1* was normalized to *Eef*. Primer-based RT-qPCR reactions were prepared in Fast SYBR Green Master Mix (Thermo Fisher Scientific). *Bcl6* qPCR primers were designed using template NM_001348026.1, Forward primer: CGCGAACCTTGATC TCCAGT, Reverse primer: CAGGGACCTGTTCACGAGAT. *Hprt* qPCR primers were designed using template NM_013556.2, Forward primer: CAGTCCCAGCGTCGTGATTA, Reverse primer: TCGAG-CAAGTCTTTCAGTCCT

## Whole skin explant culture

Embryonic back skin was dissected from E13.0 - E14.25 embryos as indicated and cultured in a Trowell-type tissue culture setup (liquid-air interface) as previously described (*Närhi and Thesleff, 2010*). For dermis cultures, back skins were obtained after treatment with Pancreatin-Trypsin or

Dispase II as described above. The culture medium (DMEM, 1X Glutamax, 10% FBS, 1% penicillin-streptomycin) was supplemented where indicated with 20 µM SU5402 (Calbiochem), 600 nM or 1.5 µM BGJ398 (Selleckchem.com, Houston, TX), 50 nM or 250 nM XL184 (Selleckchem.com, Houston, TX), or with 0.1% DMSO only. Explants were fixed in 4% PFA for immunostaining (see below).

## FGF bead treatment of embryonic dermises

Heparin-agarose beads (Ø=70–100 µm, MCLAB, San Francisco, CA) were loaded with 100 µg/ml FGF9 or FGF20 or 0.1% BSA for 2 hr at room temperature and subsequently washed with PBS. Dermises from E13.0–13.5 wild-type NMRI embryos were pooled for gene expression analysis, cell shape, and EdU incorporation assay. Dermises from E13.0 – E13.5 $Fgf20^{-/-}$ embryos were used in pairs for density analysis to compare treatment with control.

## EdU incorporation

To label proliferating cells, pregnant dams mated with $Fgf20^{-/-}$ males were given one intraperitoneal injection of 25 mg/kg body weight EdU (Life Technologies, Eugene, OR) dissolved in saline 2 hr prior to sacrifice. To label cultured dermises,NMRI E13.5 dermises cultured with FGF20- or FGF9-loaded beads for 18 hr as described above. During the last 2 hr, 10 µM EdU (Life Technologies, Eugene, OR) was introduced in the medium. The samples were then fixed with 4% PFA and EdU detection was performed with Click-iT kit (Life Technologies, Eugene, OR) according to manufacturer's protocol. Briefly, samples were permeabilized with 3% BSA, 0.5% Triton X-100 (MP Biomedicals, Solon, OH) for 1 hr, stained with Click-iT reaction cocktail containing Alexa488-azide for 2 hr protected from light, washed thoroughly for 2 hr with PBS and mounted in Vectashield and imaged using Lumar.V12 stereomicroscope with 1.2x objective and AxioCam ICc camera (all Zeiss).

## Immunostaining and whole mount confocal microscopy

The following antibodies and reagents were used: Sox2 (goat polyclonal, Santa Cruz, SC-17320, Dallas, TX) 1:500 for sections and 1:200 for whole-mounts, Sox2 (rabbit polyclonal, Stemgent, 09–0024, Glasgow, UK) 1:300 for whole-mounts, β-galactosidase (β-gal, rabbit polyclonal, MP Biomedicals, 55976, Solon, OH) 1:1500, β-galactosidase (β-gal, chicken polyclonal, Abcam, ab9361, Cambridge, UK) 1:1500, EpCAM (rat monoclonal, BD Pharmingen, 552370, San Diego, CA) 1:500, Keratin 14 (rabbit monoclonal, Thermo Fisher Scientific, RB-9020-P, Runcorn, UK) 1:500, and Cdkn1a (rabbit monoclonal, Abcam, ab188224, Cambridge, UK) 1:1000. Alexa 488, 568, or 647-conjugated secondary antibodies (Life Technologies) were used at 1:500 for sections and 1:400 for whole-mount samples for staining. For labeling of multiple antigens, the primary antibodies and secondary antibodies were incubated simultaneously.

For immunostaining tissue sections, microscope slides were deparaffinized and washed with PBS for 10 min. Antigen retrieval was performed by incubation with 10 mM Na-citrate acid at 100°C for 10 min and the sections were allowed to cool to room temperature slowly. For fluorescent labeling, sections were permeabilized with 0.1% Triton X-100, 10 min and blocked with 10% normal donkey serum, 0.1% Triton X-100 for 30 min. The sections were stained with primary antibodies overnight at 4°C and washed with PBS at room temperature. Slides were incubated with Alexa Fluor-conjugated secondary antibodies at room temperature for 2 hr, washed with PBS, and mounted with Vectashield containing DAPI (Vector Laboratories, Burlingame, CA). For immunohistochemical labeling, slides were stained using BrightVision Poly-HRP-AntiRB-kit (DVPR110HRP, ImmunoLogic, Duiven, The Netherlands) according to manufacturer's instructions. Briefly, slides were blocked using Pre-Antibody Blocking solution, 5 min, RT and washed with PBS. Then primary antibody was diluted in Pre-Antibody Blocking and the slides were stained overnight, 4°C, before washing with PBS. The slides were then stained with secondary goat, anti-rabbit-Poly-HRP antibody 30 min, RT and washed with PBS. The slides were then treated with DAB-solution (BS04-110, ImmunoLogic, Duiven, The Netherlands) according to maunfacturer's instructions, 8 min, RT, washed in de-ionized water, and mounted with ImmuMount (ThermoScientific, Kalamazoo, MI). Images were acquired with Axio Imager M.2 microscope equipped with Plan-Neofluar 20x/0.5 objective and AxioCam HRc.

For whole-mount confocal microscopy, embryonic skin was spread onto 0.1 µm nucleopore filters and fixed for 2 hr at room temperature. Tissues were washed in large volumes of PBS and subsequently blocked in 10% normal donkey serum, 0.4% Triton X-100 for 1 hr, 4°C. Blocking solution was

changed directly for primary antibody incubation in blocking solution 12–48 hr. The samples were washed in several changes of large volumes of PBS over 6–24 hr. Secondary antibody and Hoechst33342 (Life Technologies, Eugene, OR) were incubated for 6–24 hr, followed by washing. In the case of Fucci reporter samples and cell shape analysis of DC and IF cells, given the differential localization of Sox2 (nuclear) and Fgf20-β-Gal (cytosolic, epithelial) we used the same secondary Alexa fluor in order to reduce imaging time. Otherwise, different fluorophores were used. Optical sections of skin were obtained using Leica TCS SP5 confocal microscope equipped with HCX APO 63x/1.30 glycerol-immersion objective (Leica, Wetzlar, Germany) at 0.5 μm intervals for bead-treated dermis and for confocal imaging of in vivo cell shape analysis using Zeiss LSM 780 confocal microscope equipped with 63x/1.40 Plan-Apochromat oil-immersion objective at 0.5 μm intervals. Otherwise, images were acquired using an upright laser scanning confocal microscope LSM700 Axio Imager.M2 (Zeiss) using LCI Plan-Neofluar 63x/1.30 glycerol-immersion objective at 0.4–0.5 μm intervals and using Plan-Apochromat 10X/0.45 air objective at 0.8 μm intervals.

## Image analysis

Z-stacks were analyzed in 3D using Imaris software (Bitplane, Zurich, Switzerland), unless stated otherwise. To analyze the number of Sox2-positive cells, co-localization of Fucci and R26R-tomato reporters and their distances to each other and the placode, surfaces of Sox2 nuclei were generated based on Sox2 immunostaining using local intensity measures. The parameters were as follows: surface level detail 0.5 μm, local contrast background subtraction (seed point diameter 4.25 μm). In the analyses of interfollicular cells, sub-placodal cells of the $Fgf20^{-/-}$ skin, and the quantification of cell shapes, Hoechst 33342 staining was exploited for surface rendering as above (surface level detail 0.6 μm, local background subtraction (seed point diameter 5 μm). To generate surfaces based on cytoplasmic GFP in Sox2-GFP cells, surface detail was 0.6 μm with local background substraction (seed point diameter 8 μm). All nuclear surfaces were manually corrected. In all analyses, center of nuclear surface was used as a marker for cell position. For reporter studies, average intensity of the reporter channel within Sox2 or Hoechst 33342 surfaces was used as a measure for a cell's reporter activity. A cut-off value was manually determined, where a cell could be distinctly identified as reporter positive. To determine the distance from placode, a placodal surface was created as above, with surface level detail of 1.5 μm.

In description of DC morphogenesis and analysis of Fgfr inhibition on DC cells, number of Sox2 + cells and their distances from the placode were determined from $Fgf20^{+/-}$ and $Fgf20^{-/-}$ skin samples immunostained for β-gal and Sox2. Cell density of the DC was determined from $Sox2\text{-}EGFP;$ $Fgf20^{+/-}$ as the number of Sox2+ surfaces inside the area distinguished by Sox2-EGFP reporter surface (created as above), the density of the interfollicular cells and sub-placodal cells of the $Fgf20^{-/-}$ samples was determined as the number of Hoechst33342 surfaces within a corresponding volume of the interfollicular tissue. Density of sub-placodal dermal cells of the $Fgf20^{-/-}$ skin was determined as the number of Hoechst33342 surfaces underneath a manually adjusted region of interest underlying the placode (β-gal). F-actin intensities in the DC and in the interfollicular dermis were determined from $Fgf20^{+/-}$ skins mount skin samples stained with A568-phalloidin (LifeTechnologies, Eugene, OR) and antibodies against Sox2 and β-gal. A surface was drawn around the Sox2-positive cells beneath placode marked by β-gal and copied to a Sox2-negative interfollicular area of the dermis at approximately the same depth to measure the mean intensity of A568-phalloidin staining. F-actin intensity in DC Sox2-positive cells and non-DC Sox2-positive cells in relation to Sox2-negative dermal fibroblasts was determined from E13.75 $Sox2\text{-}EGFP;Fgf20^{+/-}$ back skin samples stained with A568-phalloidin and β-gal antibody. Sox2-GFP surfaces demarcating individual cells were classified as DC or non-DC based on the proximity to the placode and mean F-actin intensities inside the surfaces measured. Dermal fibroblast surfaces were drawn based on phalloidin-staining at locations close to the non-DC Sox2-GFP cells. Distance of Sox2+ surfaces to placode, were measured as the shortest distance to β-gal surface. Distance to nearest neighbor was determined based on Sox2 surface center coordinated using R software and nabor package (version 0.4.7) and median distances of each cell group within each placode were analyzed using Mann-Whitney test. For lineage tracing analysis, $Sox2^{CreERT/+};R26R^{tdTomato/+};Fgf20^{+/-}$ skin immunostained for Sox2 and β-gal were used. DC cell number, tdTomato positivity and position were determined by Sox2 surfaces as described above. Distance to placode was determined as the distance to a manually designated point on the center of placode surface and the identity of the nearest neighbor was determined based on Sox2 surface

coordinates using R software with nabor package (version 0.4.7). For cell cycle analysis, confocal stacks of $Fgf20^{+/-};Fucci$ reporter skins immunostained for Sox2 and β-gal were used. Sox2 surfaces were used for DC cells and Hoechst33342 surfaces were used for interfollicular cells of $Fgf20^{+/-}$ skin and sub-placodal cells of the $Fgf20^{-/-}$ skin. Reporter positivity was determined as above.

DC and IF nuclear shapes were determined from Hoechst33342 staining using Imaris software sphericity measurement. Non-adjacent cells were selected for analysis in order to avoid bias introduction from manual surface editing. IF control cells were selected at ~70 μm distance from condensate center. Cell shape was also determined using $Fgf20^{+/-};R26R^{mTmG}$ skins immunostained for Sox2 and β-gal. Cell shapes were analyzed with ImageJ software based on membrane-bound tdTomato signal with rolling ball background subtraction (5 μm diameter), 3D Gaussian filtering (0.4 μm diameter) and Gamma correction (0.6 value). The mT-negative region containing the cytoplasm was segmented using the Wand-tracing tool in the Segmentation Editor of Image J. The generated objects were automatically detected with the 3D Objects Counter, exported into the 3D ROI Manager (*Ollion et al., 2013*) and visualized/animated with the 3D Viewer.

For studies with FGF-loaded beads, $Fgf20^{-/-}$ dermises were used. A region within a 30 μm distance from the bead surface at its midsection was analyzed from confocal stacks for cell density and nuclear shape. Cell density was measured by manual counting from optical sections and nuclear shapes were derived from Hoechst33342 surfaces. Nuclear surfaces of cells surrounding the middle third of the bead height were used for analysis and cells at 0–15 μm and 15–30 μm distances from the bead surface were analyzed.

## Live confocal imaging and image analysis

E13.5 *Sox2-EGFP;Fucci-mKO* back skins were explanted into Trowell-type culture with DMEM/F12 (no phenol red), 10% FBS, 1% penicillin-streptomycin as previously described (*Ahtiainen et al., 2014*). Briefly, explants were allowed to recover a minimum of 2 hr after dissection and then imaged with Leica SP5 laser scanning confocal microscope using HC PL APO 10x/0.4 objective and equipped with an incubation chamber (LifeImagingServices) to maintain 5% $CO_2$ humidified atmosphere at 37°C. Confocal image stacks were acquired at 3.25 μm intervals with 10% laser power, 700 Hz scanning speed, and sub-optimal sampling every 20 min to minimize laser-induced damage.

Time-lapse videos were analyzed using Imaris software. Tissue drift was corrected by manually tracing a minimum of seven non-motile cells over time and using the software's translational drift correction algorithm. Sox2-GFP+ cell movements were manually traced back in time based on Fucci-mKO signal for as long as cell tracking could be reliably done. DC center was determined as the center of the Sox2-GFP signal of the DC. Similar tracking was performed on interfollicular dermal fibroblasts around an arbitrary migration center. Variables of cell movement were as follows:

Velocity = track length/track duration
Net velocity = track displacement length/track duration
Track straightness = displacement length/track length

$$cos\alpha = \frac{\bar{a}\bar{b}}{|\bar{a}| \cdot |\bar{b}|}$$

where α is the escape angle, $\bar{a}$ is cell trajectory, and $\bar{b}$ is a vector between cell's starting position and DC center or migration center for DC and IF cells, respectively.

## Statistical testing

The normality of distributions of data were analyzed using the Shapiro-Wilk test (confidence interval 95%). Normally-distributed data was analyzed using two-tailed Student's T-test or One-Way Anova, while a non-parametric Mann-Whitney U test was performed to analyze statistical difference of data that failed Shapiro-Wilk test. When data was normalized, a one-sample T-test was performed. $\chi^2$-test was used to analyze randomness of cell neighbor distribution in lineage-tracing analysis. Rayleigh's Z-test was conducted to test normal distribution of cell movement direction, followed by Watson's U2 test to analyze significance of cell movement direction distributions in live imaging experiments.

## Acknowledgements

We thank Ms. Raija Savolainen, Ms. Riikka Santalahti, Ms. Merja Mäkinen, and Ms. Agnes Viherä for excellent technical assistance, and Jukka Jernvall for support and the *Fucci* mice provided by the RIKEN BioResource Center through the National Bio-Resource Project of the MEXT, Ibaraki, Japan. Laura Ahtiainen is acknowledged for help in setting up the time-lapse imaging, and the Mikkola lab for helpful discussions. Confocal imaging was conducted at the Light Microscopy Unit of Institute of Biotechnology, University of Helsinki, and transmission electron microscopy was completed in the Electron Microscopy Unit at the Institute of Biotechnology, University of Helsinki. RNAsequencing was performed in the DNA Sequencing and Genomics Unit at the Institute of Biotechnology, University of Helsinki. This work was funded by the Academy of Finland, the Sigrid Jusélius Foundation, Jane and Aatos Erkko Foundation, and the Doctoral Program in Integrative Life Science of the University of Helsinki.

## Additional information

### Funding

| Funder | Grant reference number | Author |
| --- | --- | --- |
| Sigrid Juséliuksen Säätiö | | Marja L Mikkola |
| Jane ja Aatos Erkon Säätiö | | Marja L Mikkola |
| University of Helsinki | Doctoral Program in Integrative Life Science | Otto JM Mäkelä |
| Academy of Finland | 268798 | Marja L Mikkola |
| Academy of Finland | 307421 | Marja L Mikkola |

The funders had no role in study design, data collection and interpretation, or the decision to submit the work for publication.

### Author contributions

Leah C Biggs, Otto JM Mäkelä, Conceptualization, Data curation, Formal analysis, Validation, Investigation, Visualization, Methodology, Writing—original draft, Project administration, Writing—review and editing; Satu-Marja Myllymäki, Conceptualization, Data curation, Formal analysis, Validation, Investigation, Visualization, Methodology, Writing—original draft, Writing—review and editing; Rishi Das Roy, Data curation, Formal analysis, Writing—original draft, Writing—review and editing; Katja Närhi, Conceptualization, Investigation, Writing—review and editing; Johanna Pispa, Tuija Mustonen, Investigation, Methodology, Writing—review and editing; Marja L Mikkola, Conceptualization, Supervision, Funding acquisition, Writing—original draft, Project administration, Writing—review and editing

### Author ORCIDs

Leah C Biggs ⬡ https://orcid.org/0000-0002-4990-8664
Otto JM Mäkelä ⬡ http://orcid.org/0000-0001-6852-9814
Rishi Das Roy ⬡ http://orcid.org/0000-0002-3276-7279
Tuija Mustonen ⬡ http://orcid.org/0000-0002-2429-5064
Marja L Mikkola ⬡ http://orcid.org/0000-0002-9890-3835

### Ethics

Animal experimentation: All mouse studies were approved and carried out in accordance with the guidelines of the Finnish national animal experimentation board under licenses KEK16-021 and ESAV/2363/04.10.07/2017.

### Decision letter and Author response

Decision letter https://doi.org/10.7554/eLife.36468.043

Author response https://doi.org/10.7554/eLife.36468.044

## Additional files

### Supplementary files
• Supplementary file 1.
DOI: https://doi.org/10.7554/eLife.36468.036
• Reporting standard 1
DOI: https://doi.org/10.7554/eLife.36468.037
• Transparent reporting form
DOI: https://doi.org/10.7554/eLife.36468.038

### Data availability
Sequencing data have been deposited in GEO under accession code GSE110459. All data analyzed for this study are included in the manuscript and supporting files. Source data files have been provided where appropriate.

The following dataset was generated:

| Author(s) | Year | Dataset title | Dataset URL | Database, license, and accessibility information |
|---|---|---|---|---|
| Biggs LC, Mäkela OJM, Myllymäki SM, Das Roy R, Närhi K, Pispa J, Mustonen T, Mikkola ML | 2018 | Identification of transcriptional targets of FGF20. | https://www.ncbi.nlm.nih.gov/geo/query/acc.cgi?acc=GSE110459 | Publicly available at the NCBI Gene Expression Omnibus (accession no. GSE110459) |

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
