## [Decision Letter]

Thank you for submitting your article "Hair follicle dermal condensation forms via *Fgf20* primed cell cycle exit, cell motility, and aggregation" for consideration by *eLife*. Your article has been reviewed by three peer reviewers, and the evaluation has been overseen by a Reviewing Editor and Randy Schekman as the Senior Editor. The reviewers have opted to remain anonymous.

The reviewers have discussed the reviews with one another and the Reviewing Editor has drafted this decision to help you prepare a revised submission.

Summary:

This manuscript is very innovative and important report arising from the expert laboratory well-known by a number of excellent publications on mechanisms regulating the development of skin and its appendages. The manuscript contains a set of fundamental data shedding light on the mechanisms underlying the formation of mesenchymal component of the hair follicle – the follicular papilla. The authors demonstrate for the first time that mesenchymal cell condensate beneath the hair follicle placode is formed through directional migration of mesenchymal cells, which establish the follicular papilla fate via exiting the cell cycle and induction of a number of dermal condensate marker genes. The manuscript is excellently written and illustrated, data presented are solid and conclusions are well justified.

Essential revisions:

1) Introduction and Discussion. p75 kD neurotrophin receptor needs to be included into the list of early markers of mesenchymal cell condensate, and the corresponding manuscript (PMID: 10588868) needs to be cited. Actually, p75NTR ablation result in upregulation of FGFR2 in the follicular papilla, thus providing a functional link between p75NTR and FGF signaling in mesenchymal cells.

2) Figure 2. To better substantiate the lack of potential contribution of Sox2^+^ Schwann cells to mesenchymal condensate formation, the authors might wish to cite the data showing the lack of Schwann cells associated with nerve fibers in the E14.5 mouse dermis (PMID: 12012374).

3) Figures 4 and 6. The migration of epithelial cells is associated with increase of phalloidin immunostaining of endogenous actin filament network. Is there any chance to demonstrate such a network in fibroblasts participating in a formation of mesenchymal cell condensate?

4) The number of mice used for each experiment is not clearly reported (except for RNAseq analysis). Multiple placodes should be examined from multiple mice to account for variation between animals. Given the low number of total placodes examined, it seems likely that only a single animal was used for many experiments. Minor point. For most experiments, It was unclear as to the number of embryos data sets came from. In particular in Figure 4, the number of cells tracked seems low and it is unclear whether multiple follicles were tracked. Please provide n values for # cells in #placodes and #embryos.

5) Since the original Fucci color contrast is not as good as later versions of the mouse, an additional method should be used to validate cell cycle exit (such as phosphohistone H3 staining or a short chase after thymidine analogue administration).

6) Since SU5402 is a broad inhibitor (VEGFR2, FGFR1, PDGFRB), other controls are needed to demonstrate that the results from skin explant studies aren't a result of Pdgf or Vegf signaling or from other FGF molecules, especially because the% of Sox2^+^ cells was decreased in response to SU5402 treatment.

7) An unresolved question is the timing and order of events in DC formation. From the data shown in Figures 2 and 3, it seems as though fibroblasts first move into the DC region via migration/aggregation, then turn on Sox2-expression and exit the cell cycle. Or do they exit the cell cycle, begin to migrate and then turn on Sox2 expression? Although the authors emphasize the advantage of using live imaging to document DC formation, it is only utilized in one experiment (Figure 4), but it would have allowed one to established the order of events in the process.

8) Figure 6 shows that addition of FGF to cultured fibroblasts is sufficient to increase the speed of migration, but whether this is true in vivo is unknown. The bead experiments are somewhat inconclusive – while beads coated with FGF are sufficient to induce expression of some target genes, the effect is too short-lived to monitor migration.

It appears that Fgf20 is sufficient to induce changes in cell density and cell shape in cells located adjacent to the bead. I'm concerned about the different bead sizes in control and experimental samples shown in Figure 6D and 6F – it seems bead circumference/surface area could affect local density and cell shape. Also, I believe the labels in Figure 6E are flipped. BSA has more cells/µm^2^ than Fgf20according to the graph but the conclusion is that Fgf20increases the cell density.

9) Fgf20 in cell cycle exit. The data nicely show that Fgf20is required for p21 expression in vivo, but is it sufficient? If Fgf20-coated beads are embedded in Fucci dermis, is this sufficient to cause cell cycle exit?

---

## [Author Response]

Essential revisions:1) Introduction and Discussion. p75 kD neurotrophin receptor needs to be included into the list of early markers of mesenchymal cell condensate, and the corresponding manuscript (PMID: 10588868) needs to be cited. Actually, p75NTR ablation result in upregulation of FGFR2 in the follicular papilla, thus providing a functional link between p75NTR and FGF signaling in mesenchymal cells.

Thank you for the suggestion. We have mentioned p75NTR in the list of early markers of the dermal condensate in the Introduction, and cited the paper (second paragraph). It is interesting to note that p75NTR deletion results in Fgfr2 upregulation, and it certainly could be that ablation of Fgf20 results in p75NTR absence and Fgfr2 upregulation, however, we did not test either of these molecules and therefore do not wish to speculate on this connection in this manuscript.

2) Figure 2. To better substantiate the lack of potential contribution of Sox2^+^ Schwann cells to mesenchymal condensate formation, the authors might wish to cite the data showing the lack of Schwann cells associated with nerve fibers in the E14.5 mouse dermis (PMID: 12012374).

We thank the reviewers for this suggestion. However, through the discovery of Fgf20 as an early marker of the hair follicle, and use of neurofilament marker 2H3 and Schwann cell precursor marker Sox2, we observe guard hair follicles already in E13.5 mouse skin with closely-associated nerve fibers decorated with faintly-stained Sox2^+^ cells (see arrows in Author response image 1, scale bar = 10 µm). This finding is in line with the RNAseq profiling of E14.5 embryonic skin (www.hair-GEL.net, PMID: 26256211) showing the presence of *Sox2*^+^ Schwann cells at the time of primary hair follicle induction.

3) Figures 4 and 6. The migration of epithelial cells is associated with increase of phalloidin immunostaining of endogenous actin filament network. Is there any chance to demonstrate such a network in fibroblasts participating in a formation of mesenchymal cell condensate?

This is an excellent point. We have used Sox2-GFP to identify both DC cells and migrating DC cells and compared the actin network of those cells outside the DC to those non-DC fibroblasts. These data have been added as Figure 4—figure supplement 2A, B. We observed an increase in the intensity of F-actin staining in the Sox2^+^ cells outside the DC when compared to Sox2- non-DC fibroblasts, likely indicating their migratory status. Interestingly, we observed a yet higher intensity of F-actin staining in the Sox2^+^ DC cells. We have further compared the intensity of phalloidin staining in the DC and compared it to the adjacent non-DC mesenchyme, and found that it was significantly increased in the DC at all stages examined in this study (new Figure 4—figure supplement 2C, D). In our analysis of live tissue, these cells showed reduced mobility (Figure 4D, F) and it is a likely possibility that F-actin in these cells is involved in maintaining the 3D structure of the DC.

4) The number of mice used for each experiment is not clearly reported (except for RNAseq analysis). Multiple placodes should be examined from multiple mice to account for variation between animals. Given the low number of total placodes examined, it seems likely that only a single animal was used for many experiments. Minor point. For most experiments, It was unclear as to the number of embryos data sets came from. In particular in Figure 4, the number of cells tracked seems low and it is unclear whether multiple follicles were tracked. Please provide n values for # cells in #placodes and #embryos.

We agree that this could be a concern and we apologize for not reporting the samples sizes in sufficient details. We have used multiple embryos in each experiment. We have provided the number of embryos, number of placodes and number of cells used in each figure legend.

5) Since the original Fucci color contrast is not as good as later versions of the mouse, an additional method should be used to validate cell cycle exit (such as phosphohistone H3 staining or a short chase after thymidine analogue administration).

We have examined the Fucci2a model (Mort et al., 2014), a second generation of cell cycle indicator mouse, during DC morphogenesis and find the results concordant with those obtained with the original Fucci strain (subsection “Dermal condensate formation is associated with cell cycle exit”, first paragraph, and representative figures are shown in the new Figure 3—figure supplement 1). In addition, we have conducted 2h EdU pulses and quantified the percentage of EdU^+^ cells in the Sox2^+^ DC cell population. In placode stage I, on average 18% of Sox2^+^ cells were EdU^+^, whereas at placode stages II-IV, very few EdU^+^ cells were observed, in line with the Fucci data. The EdU data have been added to Figure 3 as panels F and G and in the Results (in the last paragraph of the aforementioned subsection).

6) Since SU5402 is a broad inhibitor (VEGFR2, FGFR1, PDGFRB), other controls are needed to demonstrate that the results from skin explant studies aren't a result of Pdgf or Vegf signaling or from other FGF molecules, especially because the% of Sox2^+^ cells was decreased in response to SU5402 treatment.

We thank the reviewers for bringing up this point. The half maximal inhibitory concentration (IC_50_) values for SU5402 areas indicated in the table below. Based on our experience higher concentrations are needed for receptor inhibition in organ culture experiments. We used SU5402 at 20 µM to achieve the reported effect; at 10 µM we did not observe any effect on placode/DC formation (n= nine skins) (data not shown). The IC_50_ values indicate that PDGFRβ inhibition is achieved with 17x greater concentration of SU5402 than FGFR inhibition. Further, conditional deletion of both PDGFRβ and PDGFRα in the developing dermis and specifically in the dermal condensate during hair follicle morphogenesis does not result in a phenotype (Rezza et al., 2015). Thus, we find it unlikely that the results would be caused by inhibition of Pdgf signaling.

We agree that the potential inhibition of Vegf signaling is a concern. Therefore, we utilized an inhibitor that is more specific to FGFR (BGJ398; see table below for IC_50_ values) and found that similarly to SU5402, in E13.5 explants treated for 24h, no DCs developed and the placode pattern resembled the Fgf20-KO model. Furthermore, we used a VEGFR specific inhibitor (Cabozantinib, XL184). At a concentration equivalent to 20 µM SU5402 (based on IC_50_ values), i.e., 50 nM, we found no effect on the DC formation or the epithelial pattern. Even the 5x higher concentration (250 nM) did not affect placode/DC formation. IC_50_ values of the inhibitors and representative figures of these data are now available as Figure 7—figure supplements 2 and 3, respectively. Based on these data we conclude that the effect of SU5402 is due to inhibition of Fgfr signaling and provide confirmation of a role for FGF signaling in DC morphogenesis.

ReagentFGFR1 IC_50_VEGFR2 IC_50_PDGFRβ IC_50_SU540230 nM20 nM510 nMBGJ3980.9 nM180 nMNot providedCabozantinib5.294 µM0.035 nMNot provided

7) An unresolved question is the timing and order of events in DC formation. From the data shown in Figures2 and 3, it seems as though fibroblasts first move into the DC region via migration/aggregation, then turn on Sox2-expression and exit the cell cycle. Or do they exit the cell cycle, begin to migrate and then turn on Sox2 expression? Although the authors emphasize the advantage of using live imaging to document DC formation, it is only utilized in one experiment (Figure 4), but it would have allowed one to established the order of events in the process.

We agree that this is a very interesting question. Our initial pilot time-lapse experiments utilized Fucci G_1_, Fucci S/G_2_/M, and Sox2-GFP transgenes, however, we did not feel as though we could reliably distinguish the Sox2-GFP from the Azami Green in Fucci S/G_2_/M, in particular in cells expressing low levels of Sox2-GFP. Due to these technical challenges and given that these tissues are transgenic reporters, we are reluctant to make any strong conclusions regarding the order of events based on our time-lapse data.

To address this point, we have compared the percentage of Fucci-G_1_ DC cells (i.e. Sox2-positive cells) in advancing placode stages (Figure 3B), and find that significantly fewer Sox2^+^ DC cells are Fucci-G1 positive at placode stage I compared to later placode stages (p_I vs II_ = 0.0211; p_II vs III_ = 0.8303, p_III vs IV_ = 0.1579). Further, our new EdU analysis in Sox2 expressing DC cells show similar results (new Figure 3F and G). Based on these data, we conclude that DC fate acquisition (assessed by Sox2 positivity) precedes cell cycle exit. We have added one sentence on this in Results, “Further, the percent non-proliferating cells in the stage I DC was significantly lower than the following stages (p_IvsII_ = 0.0211), suggesting that DC fate acquisition occurs before cell cycle exit.” We hope to address this important question in future studies with more sophisticated tools.

8) Figure 6 shows that addition of FGF to cultured fibroblasts is sufficient to increase the speed of migration, but whether this is true in vivo is unknown. The bead experiments are somewhat inconclusive – while beads coated with FGF are sufficient to induce expression of some target genes, the effect is too short-lived to monitor migration.It appears that Fgf20 is sufficient to induce changes in cell density and cell shape in cells located adjacent to the bead. I'm concerned about the different bead sizes in control and experimental samples shown in Figure 6D and 6F – it seems bead circumference/surface area could affect local density and cell shape. Also, I believe the labels in Figure 6E are flipped. BSA has more cells/µm2 than Fgf20 according to the graph but the conclusion is that Fgf20 increases the cell density.

We apologize for not making it clear that the size of the beads used in the experiment vary from 70 to 100 µm in diameter; we have now added this information in Materials and methods. Beads were randomly chosen for each experiment. We have retrospectively measured the diameter of all beads used in this experiment and we find no statistical difference between the FGF20- or FGF9-loaded beads and their controls. Bead diameters were (AVR ± S.D.): FGF20 = 87 ± 10µm, BSA (control for FGF20) = 85 ± 10µm, FGF9 = 87 ± 7µm and BSA (control for FGF9) = 86 ± 11µm. According to a paired two-tailed Student’s T-test, FGF20- and FGF9-loaded beads were not significantly different from their BSA controls (P^FGF20-BSA^ = 0.736 and P^FGF9-BSA^ = 0.923; n ^FGF20-BSA^ = 9 pairs, n ^FGF9-BSA^ = 7 pairs). We have changed the image in Figure 6D and F to better match the average size of the beads and to avoid any misunderstandings.

We apologize for the flip of the labels in Figure 6E. We have corrected the error.

9) Fgf20 in cell cycle exit. The data nicely show that Fgf20 is required for p21 expression *in vivo*, but is it sufficient? If Fgf20-coated beads are embedded in Fucci dermis, is this sufficient to cause cell cycle exit?

To address this issue, we have performed additional 3h bead experiments on intact mesenchyme. Using p21 antibody, we did not observe any cells next to the bead that were positive. In addition to whole mount in situ hybridization, we also used a ^35^S-labeled probe in sections and found no induction of p21 around FGF20 or FGF9-coated beads. No significant effect could be detected in percentage of Fucci-G1+ cells around FGF20 or FGF9 bead either. Thus, it appears that Fgf20 alone is insufficient to induce p21 and cell cycle exit and may require additional cues. Alternatively, the kinetics of p21 induction may be slower than that of the pathway feedback inhibitors such as Spry4 and Dusp6.